# On-chip bacterial foraging training in silicon photonic circuits for projection-enabled nonlinear classification

Guangwei Cong [1✉], Noritsugu Yamamoto[1], Takashi Inoue [1], Yuriko Maegami [1], Morifumi Ohno[1], Shota Kita[2], Shu Namiki [1] & Koji Yamada [1]

On-chip training remains a challenging issue for photonic devices to implement machine learning algorithms. Most demonstrations only implement inference in photonics for offline-trained neural network models. On the other hand, artificial neural networks are one of the most deployed algorithms, while other machine learning algorithms such as supporting vector machine (SVM) remain unexplored in photonics. Here, inspired by SVM, we propose to implement projection-based classification principle by constructing nonlinear mapping functions in silicon photonic circuits and experimentally demonstrate on-chip bacterial foraging training for this principle to realize single Boolean logics, combinational Boolean logics, and Iris classification with ~96.7 − 98.3 per cent accuracy. This approach can offer comparable performances to artificial neural networks for various benchmarks even with smaller scales and without leveraging traditional activation functions, showing scalability advantage. Natural-intelligence-inspired bacterial foraging offers efficient and robust on-chip training, and this work paves a way for photonic circuits to perform nonlinear classification.

[1] Platform Photonics Research Center, National Institute of Advanced Industrial Science and Technology (AIST), 16-1, Onogawa, Tsukuba, Ibaraki 305-8569, Japan. [2] NTT Basic Research labs., 3-1, Morinosato Wakamiya, Atsugi-shi, Kanagawa 243-0198, Japan. ✉email: gw-cong@aist.go.jp

I mplementing machine learning algorithms in optical means is gaining wide interest[1–4]. Because light propagation in physical systems can execute various mathematical processes required by these algorithms, such as vector-matrix multiplication (VMM)[5–8], random projection[9,10], and Fourier transformation[11], optical and photonic systems can offer better energy efficiency and higher speed than electronic ones and thus serve as an alternative hardware for offloading tasks from electronic processors to mitigate the rapidly increased burden on computing resources associated with wide deployment of machine learning algorithms. As a general notable approach to machine learning, artificial neural networks (ANNs) are one of the most optically deployed algorithms. To date, experimental demonstrations of optical neural networks (ONNs) include free-space optics[10–12], fiber-based system[7], and integrated photonics[5,6,8,13,14]. Each of them has pros and cons in respect of integrability, system complexity, scalability, while most demonstrations still only perform inference in optics/photonics by applying knowledge obtained from offline-trained neural network models. This is acceptable for a task-specialized hardware to execute a fixed task, however, training always relying on another duplicate neural network program on computer will forfeit the advantages[15] promised by utilizing optics/photonics and make it difficult for ONNs to be standalone, especially when real-time reconfiguration is required for changing tasks. Integrated photonic circuits have advantages in high integration degree and reconfigurability compared to other platforms, but simultaneous on-chip training and inference are still less demonstrated in experiment[16]. On-chip training in experiment remains a challenging issue for photonic devices to implement machine learning algorithms. The difficulty lies in that the photonic chips cannot be directly trained by the gradient-based backpropagation (BP) algorithm widely used in ANN programs. This algorithm is highly efficient in that the gradients of both linear and nonlinear layers can be updated directly using their output values, however, there is lack of an efficient protocol for photonic devices to update the gradient information in electrical domain for phase shifters (PS) from errors in optical domain. In theory, BP-based optical training[15,17] and offline-calculation-assisted parallel calibration methods[18] were proposed, which are feasible provided taking additional measures like utilizing locally embedded monitors and amplifier circuits for local optical-to-electrical feedback and preparing optical feedback vectors. Thus, applying the weights trained by an offline model to pre-calibrated photonic devices directly or via matrix decomposition remains a mainstream approach to demonstrate an inference-only ONN in experiment.

For on-chip training, global optimization algorithms[16,19] and forward propagation (FP) algorithms[6] (forward differential method to evaluate the gradient for each element by two-times propagation) are two practical ways if only replying on the output detectors, despite that the latter one has not been demonstrated yet in actual experiments, as far as we know. Recently, on-chip training a photonic chip using genetic algorithms was experimentally demonstrated[16]. We also experimentally demonstrated bacterial foraging optimization (BFO), a natural intelligence inspired algorithm, for on-chip training silicon photonic circuits to realize real-time reconfiguration and failure recovery[20] and XOR (exclusive OR) separation[21]. It is well-known that the gradient-based algorithms no matter BP or FP sometime experience gradient vanishing or exploding and local minima problems, resulting in failures of training, while the BFO algorithm emulating bacterial behavior in foraging nutrition is free from such problems and has better global convergence capability because a bacteria (i.e., state vector) moves along a randomly-selected direction in high dimensional space instead of gradient descent direction[22], offering a possibility to escape from local minima.

On the other hand, except for ANNs (in form of multilayer perceptron), there are other machine learning algorithms such as supporting vector machine (SVM)[23–25]. SVM is a kind of supervised machine learning algorithm, which finds wide applications in classification and regression problems and can offer high generalization capability and high performance[26]. Originally, SVM is a linear classifier which solves the maximum-margin hydroplane to separate the linearly separable data. Later, it is developed to nonlinear classifier by introducing the idea of nonlinear projection which projects the data from the input space to a higher-dimensional feature space to seek an easy linear separation[24]. Although the classifier is linear in the transformed higher-dimensional feature space, it can correspond to a nonlinear decision boundary in the original input space. In practical use of SVM on the program side, kernel method is applied so that the whole algorithm can be kept similar as the linear classifier except for replacing the dot product by a kernel function, without involving in explicit computing using the nonlinear mapping function. It is possible to explicitly create nonlinear mapping functions in hardware to implement this SVM-like idea; while to date, few efforts have been made to explore this idea in photonic platforms. Random projection using scattering media[9,10] and reservoir computing[27] using dynamics of time-delay systems could be regarded as implicit projection-based algorithms.

In this work, inspired by SVM, we propose to construct nonlinear mapping functions directly by arranging data input schemes in Mach–Zehnder interferometer (MZI) networks to realize projection-enabled nonlinear classification and fabricate silicon photonic circuits to implement this SVM-like principle for proof-of-concept demonstration. We experimentally implement both on-chip bacterial foraging training and inference in the fabricated silicon photonic chip without either doing pre-calibration for phase errors or using a pre-trained network model on computer, in a standalone mode. Using this projection-based approach, we experimentally demonstrate single Boolean logics (XOR, AND, OR, NAND), combinational Boolean logics (XOR-AND, OR-NAND) in a single photonic chip, and Iris classification with up to ~98.3% test accuracy. This approach can achieve comparable classification performance to ANNs for various benchmarks (test accuracies of 100%, 100%, ~90%, ~94% for the nonlinear datasets of 2-classes Circle, 2-classes Moon, 3-classes Spiral, and MNIST handwritten digit, respectively) even with much smaller scales and without leveraging activation functions, showing scalability advantages. This work paves a way for photonic devices to perform nonlinear classification.

## Results

**Principle and device topology**. The basic idea of SVM is depicted in Supplementary Fig. 2(a) which shows a quadratic mapping enabled linear separation. As seen, after projection, it is easy to find a line with a maximum margin to separate the binary-class data. This projection-assisted separation via directly treating the input data by a mapping function is the fundamental idea of SVM, which is usually not included in the conventional ANN frame. Similarly, we propose to utilize sinusoidal mapping to implement this idea, as shown in Supplementary Fig. 2(b), since the sinusoidal function is inherently of the phase-amplitude nonlinearity of photonic devices such as MZIs. When remaining the data in the phase space (input space in electrical domain) and taking the optical amplitude space as the transformed feature space, we can construct sinusoidal mapping functions from the phase space (real) to amplitude space (complex) in photonic circuits. Figure 1a shows the topology of projection-based photonic classifier (PPC) to be demonstrated in this work, which is consisted of MZIs and PS. For a dataset $\mathbf{D} = \{(\mathbf{x}, t) \mid \mathbf{x} \in \mathbb{R}^p,$

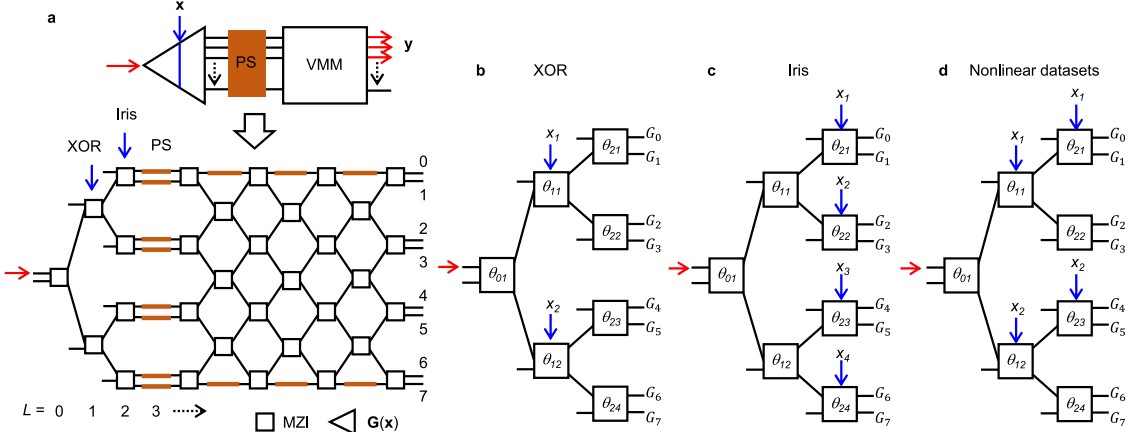

**Fig. 1 Schematics of the projection-based photonic classifier. a** Topology to be fabricated for experimental demonstration. A schematic to present the device topology is depicted at the top. The layer $L$ is numbered for each column of MZI (Mach–Zehnder interferometer) or phase shifters (PS) (in total 11 layers). Data-input schemes for constructing mapping functions $\mathbf{G}(\mathbf{x}) = \{G_0, G_1, ..., G_7\}$ by arranging the data input into the MZI-based photonic network: **b** XOR (exclusive OR); **c** Iris; **d** Nonlinear datasets, in which each MZI has a phase bias $\theta_{jk}$ ($j$: the column number, $k$: the row number) as its learning parameter. The topology in (**a**) can accommodate the schemes in (**b**–**d**) by using the layer $L = 1$ or 2 and both for data input. VMM (vector-matrix multiplication) has a 7-layer MZI mesh (column × row = 4 × 4 + 3 × 3). All MZIs use 50:50 splitters. XOR and the nonlinear datasets have two parameters $\mathbf{x} = (x_1, x_2)$ ($x_i = 0$ or 1 for XOR, $x_i \in [0,1)$ for nonlinear datasets), while the Iris dataset has four parameters $\mathbf{x} = (x_1, x_2, x_3, x_4)$. Red arrows indicate the light input and output. Blue arrows indicate the data input from electrical domain. Yellow color denotes the phase shifter.

$t \in (0, 1, 2, ..., m)\}_N$ consisting of $p$-dimensional input vectors $\mathbf{x}$ in real (R) space of $N$ samples and their corresponding labels $t$ of $m$-classes, the data $\mathbf{x}$ is input as optical phase into a cascaded MZI network to produce a mapping function $\mathbf{G}(\mathbf{x})_n$ that can project the data $\mathbf{x}$ to a $n$-dimensional space ($n \geq p$). A VMM module follows $\mathbf{G}(\mathbf{x})$ to perform the matrix transform to complete linear separation in the projected space. Executing VMM is the advantage of photonic circuits, which can be done in several ways[5–8,28]. Here we use an interference circuit based on the Clements' topology[5,28]. A column of PS is inserted between $\mathbf{G}(\mathbf{x})$ and VMM for phase tuning in real devices and for rotating the projected vector in the $n$-dimension space. In VMM, we do not use the external PS for vector rotation for all MZIs and only at the top and bottom rows we add 6 heaters (PS) for tuning phase errors in the fabricated devices. By simulation, we confirm that including the external phase shifter or not does not influence the classification results for this projection-based method (see Supplementary Section 6.3). Thus, to keep consistency, for comparison with experiment, the simulation uses the exact same structure in Fig. 1a, while for comparison between this PPC and ANN, all MZIs do not include external PS in simulations. Furthermore, the influence of layer number of VMM is also clarified in Supplementary Section 6.2. With this work as reference, adding more duplicate tunable parameters obviously will not degrade the classification results.

**Data input and device operators.** Figure 1b, c, and d show the data input schemes for XOR, Iris dataset[29], and nonlinear datasets (Circle, Moon, etc.), respectively, in which we omit the PS and VMM. Based on the MZI equation (Eq.) in Supplementary Eq. 1, each element of the mapping function $\mathbf{G}(\mathbf{x}) = \{G_0, G_1, ..., G_7\}$ can be derived as the equations (Eqs.) (1)–(8) for XOR, (9)–(16) for Iris, and (17)–(24) for nonlinear datasets.

$$G_0 = \frac{1}{2}\left(e^{i\theta_{01}} - 1\right) \cdot \frac{1}{2}i\left(e^{i(x_1+\theta_{11})} + 1\right) \cdot \frac{1}{2}i\left(e^{i\theta_{21}} + 1\right) \quad (1)$$

$$G_1 = \frac{1}{2}\left(e^{i\theta_{01}} - 1\right) \cdot \frac{1}{2}i\left(e^{i(x_1+\theta_{11})} + 1\right) \cdot \frac{1}{2}\left(1 - e^{i\theta_{21}}\right) \quad (2)$$

$$G_2 = \frac{1}{2}\left(e^{i\theta_{01}} - 1\right) \cdot \frac{1}{2}\left(1 - e^{i(x_1+\theta_{11})}\right) \cdot \frac{1}{2}\left(e^{i\theta_{22}} - 1\right) \quad (3)$$

$$G_3 = \frac{1}{2}\left(e^{i\theta_{01}} - 1\right) \cdot \frac{1}{2}\left(1 - e^{i(x_1+\theta_{11})}\right) \cdot \frac{1}{2}i\left(e^{i\theta_{22}} + 1\right) \quad (4)$$

$$G_4 = \frac{1}{2}i\left(e^{i\theta_{01}} + 1\right) \cdot \frac{1}{2}\left(e^{i(x_2+\theta_{12})} - 1\right) \cdot \frac{1}{2}i\left(e^{i\theta_{23}} + 1\right) \quad (5)$$

$$G_5 = \frac{1}{2}i\left(e^{i\theta_{01}} + 1\right) \cdot \frac{1}{2}\left(e^{i(x_2+\theta_{12})} - 1\right) \cdot \frac{1}{2}\left(1 - e^{i\theta_{23}}\right) \quad (6)$$

$$G_6 = \frac{1}{2}i\left(e^{i\theta_{01}} + 1\right) \cdot \frac{1}{2}i\left(e^{i(x_2+\theta_{12})} + 1\right) \cdot \frac{1}{2}\left(e^{i\theta_{24}} - 1\right) \quad (7)$$

$$G_7 = \frac{1}{2}i\left(e^{i\theta_{01}} + 1\right) \cdot \frac{1}{2}i\left(e^{i(x_2+\theta_{12})} + 1\right) \cdot \frac{1}{2}i\left(e^{i\theta_{24}} + 1\right) \quad (8)$$

$$G_0 = \frac{1}{2}\left(e^{i\theta_{01}} - 1\right) \cdot \frac{1}{2}i\left(e^{i\theta_{11}} + 1\right) \cdot \frac{1}{2}i\left(e^{i(x_1+\theta_{21})} + 1\right) \quad (9)$$

$$G_1 = \frac{1}{2}\left(e^{i\theta_{01}} - 1\right) \cdot \frac{1}{2}i\left(e^{i\theta_{11}} + 1\right) \cdot \frac{1}{2}\left(1 - e^{i(x_1+\theta_{21})}\right) \quad (10)$$

$$G_2 = \frac{1}{2}\left(e^{i\theta_{01}} - 1\right) \cdot \frac{1}{2}\left(1 - e^{i\theta_{11}}\right) \cdot \frac{1}{2}\left(e^{i(x_2+\theta_{22})} - 1\right) \quad (11)$$

$$G_3 = \frac{1}{2}\left(e^{i\theta_{01}} - 1\right) \cdot \frac{1}{2}\left(1 - e^{i\theta_{11}}\right) \cdot \frac{1}{2}i\left(e^{i(x_2+\theta_{22})} + 1\right) \quad (12)$$

$$G_4 = \frac{1}{2}i\left(e^{i\theta_{01}} + 1\right) \cdot \frac{1}{2}\left(e^{i\theta_{12}} - 1\right) \cdot \frac{1}{2}i\left(e^{i(x_3+\theta_{23})} + 1\right) \quad (13)$$

$$G_5 = \frac{1}{2}i\left(e^{i\theta_{01}} + 1\right) \cdot \frac{1}{2}\left(e^{i\theta_{12}} - 1\right) \cdot \frac{1}{2}\left(1 - e^{i(x_3+\theta_{23})}\right) \quad (14)$$

$$G_6 = \frac{1}{2}i\left(e^{i\theta_{01}} + 1\right) \cdot \frac{1}{2}i\left(e^{i\theta_{12}} + 1\right) \cdot \frac{1}{2}\left(e^{i(x_4+\theta_{24})} - 1\right) \quad (15)$$

$$G_7 = \frac{1}{2}i\left(e^{i\theta_{01}} + 1\right) \cdot \frac{1}{2}i\left(e^{i\theta_{12}} + 1\right) \cdot \frac{1}{2}i\left(e^{i(x_4+\theta_{24})} + 1\right) \quad (16)$$

$$G_0 = \frac{1}{2}\left(e^{i\theta_{01}} - 1\right) \cdot \frac{1}{2}i\left(e^{i(x_1+\theta_{11})} + 1\right) \cdot \frac{1}{2}i\left(e^{i(x_1+\theta_{21})} + 1\right) \quad (17)$$

$$G_1 = \frac{1}{2}\left(e^{i\theta_{01}} - 1\right) \cdot \frac{1}{2}i\left(e^{i(x_1+\theta_{11})} + 1\right) \cdot \frac{1}{2}\left(1 - e^{i(x_1+\theta_{21})}\right) \quad (18)$$

$$G_2 = \frac{1}{2}\left(e^{i\theta_{01}} - 1\right) \cdot \frac{1}{2}\left(1 - e^{i(x_1+\theta_{11})}\right) \cdot \frac{1}{2}\left(e^{i\theta_{22}} - 1\right) \quad (19)$$

$$G_3 = \frac{1}{2}\left(e^{i\theta_{01}} - 1\right) \cdot \frac{1}{2}\left(1 - e^{i(x_1+\theta_{11})}\right) \cdot \frac{1}{2}i\left(e^{i\theta_{22}} + 1\right) \quad (20)$$

$$G_4 = \frac{1}{2}i\left(e^{i\theta_{01}} + 1\right) \cdot \frac{1}{2}\left(e^{i(x_2+\theta_{12})} - 1\right) \cdot \frac{1}{2}i\left(e^{i(x_2+\theta_{23})} + 1\right) \quad (21)$$

$$G_5 = \frac{1}{2}i\left(e^{i\theta_{01}} + 1\right) \cdot \frac{1}{2}\left(e^{i(x_2+\theta_{12})} - 1\right) \cdot \frac{1}{2}\left(1 - e^{i(x_2+\theta_{23})}\right) \quad (22)$$

$$G_6 = \frac{1}{2}i\left(e^{i\theta_{01}} + 1\right) \cdot \frac{1}{2}i\left(e^{i(x_2+\theta_{12})} + 1\right) \cdot \frac{1}{2}\left(e^{i\theta_{24}} - 1\right) \quad (23)$$

$$G_7 = \frac{1}{2}i\left(e^{i\theta_{01}} + 1\right) \cdot \frac{1}{2}i\left(e^{i(x_2+\theta_{12})} + 1\right) \cdot \frac{1}{2}i\left(e^{i\theta_{24}} + 1\right) \quad (24)$$

All Eqs. (1)–(16), (19), (20), (23), and (24) can be written as the form of $G = w_j e^{ix} + b_j$ and Eqs. (17), (18), (21), and (22) as the form of $G = (w_j e^{ix} + b_j)^2$, where $w_j$ and $b_j$ ($j = 1, 2, …, n$) are complex coefficients determined by phase biases. We use a matrix **A** to present the matrix product of VMM and PS (diagonal matrix), and then, the whole device can be expressed as the Eqs. (25) and (26) for XOR and Iris, and nonlinear datasets, respectively. As shown in following sections, classifying these nonlinear datasets without using nonlinear activation functions evidences the role of nonlinear projection. For training the device, we use the optical power of output vector **y** and a loss function $\mathscr{L}$ of the mean squared error (MSE) as shown in Eq. (27), where a one-hot vector $\mathbf{y_t}$ converted from labels $t$ is used as the target vector.

$$\mathbf{y} = \mathbf{AG} = \mathbf{A}\left[\left(w_1 e^{ix_1} + b_1\right), \left(w_2 e^{ix_1} + b_2\right), … \right] \quad (25)$$

$$\mathbf{y} = \mathbf{AG} = \mathbf{A}\left[\left(w_1 e^{ix_1} + b_1\right)^2, \left(w_2 e^{ix_1} + b_2\right)^2, \left(w_3 e^{ix_1} + b_3\right), … \right] \quad (26)$$

$$\mathscr{L} = \min\sum\left(\mathbf{y_t} - |\mathbf{y}|^2\right)^2/N \quad (27)$$

**Model explanation**. Here we explain why the model proposed above can be understood as an SVM-like method in two aspects: distance maximization and kernel matrix. Once trained, the device establishes a linear equation $\mathbf{y} = \mathbf{wx'^T} + \mathbf{b}$ in the $n$-dimensional complex space, where $\mathbf{x'}$ presents the axis vector only containing the Euler's terms in the Eqs. (25) and (26). We introduce another axis in which the correct label is marked as zero and thus the coordinate vector becomes a $n+1$-dimension one $(\mathbf{x'}, 1 - \mathbf{y_t})$. This linear equation means a hydroplane in the $n+1$-dimensional complex space by changing it to $[\mathbf{w}, 1][\mathbf{x'}, -\mathbf{y}]^T + \mathbf{b} = 0$. These data coordinates are separated by this plane and their distances to the plane are $\mathbf{d} = |\mathbf{wx'^T} + 1 - \mathbf{y_t} + \mathbf{b}|/||[\mathbf{w},1]|| = |1 - \mathbf{y_t} + \mathbf{y}|/||[\mathbf{w},1]||$. For the $j$-th element, it can be easily noticed that $d_j \leq (|1 - y_{t,j}| + |y_j|)/||[\mathbf{w},1]|| \leq (1 + |y_j|)/||[\mathbf{w},1]|| \leq 2/||[\mathbf{w},1]||$ since the maximum of $|1 - y_{t,j}|$ is one and $|y_j|$ approaches $y_{t,j}$ ($|y_j| \rightarrow y_{t,j}$) in the training. On the other side, $d_j = |1 - y_{t,j} + y_j|/||[\mathbf{w},1]|| = |1 - (y_{t,j} - y_j)|/||[\mathbf{w},1]|| \geq (1 - |(y_{t,j} - y_j)|)/||[\mathbf{w},1]|| \geq (1 - (|y_{t,j}| - |y_j|))/||[\mathbf{w},1]|| = (1 - \varepsilon)/||[\mathbf{w},1]||$, where $\varepsilon$ is a residual related to the loss due to $|y_j| \rightarrow y_{t,j}$. The distances for all data points will vary in this finite region between the maximum and minimum values. The training loss determines the minimum distance which is the

maximum margin for the data points of supporting vectors (e.g., if the loss is negligible, $\varepsilon \rightarrow$ zero and then $d$ approaches $1/||[\mathbf{w},1]||$, which are the maximum margin). Therefore, minimizing the loss (Eq. 27) is to maximize the power (i.e., modulus) of complex vector **y** and the training using the power is equivalent to optimizing the distance in the high dimensional complex space.

On the other aspect, the proposed PPC can also be understood from the kernel matrix. The training is not directly involved in the kernel matrix in this work, while the inner product of the linear equation established by training between each two samples can be done to describe the sample similarity, which can produce another linear equation based on the kernel matrix. This matrix is consisted of the Hermitian inner product of **G(x)** by traversing any pair of two samples **x, v** (Eq. 28). We can derive the form of this kernel matrix from Eqs. (1)–(24) for either XOR, Iris, or nonlinear datasets. As seen, these equations are of the product of three terms and each term is a component like $C = 0.5i(e^{i\theta} + 1)$ or $C' = 0.5(1 - e^{i\theta})$. Due to $CC^* + C'C'^* = 1$, obviously, the matrix **H** has unity diagonal elements ($\mathbf{x} = \mathbf{v}$). For the inner product between different samples ($\mathbf{x} \neq \mathbf{v}$), we take the Eqs. (1)–(8) (XOR) as example to derive out the Eqs. (29)–(32) (using $w'$ to replace the omitted part for conciseness). The Eq. (31) is the sum of Eqs. (29) and (30) and then the inner product is the sum of Eqs. (31) and (32), which can be written as Eq. (28) by omitting the constant term of one. Obviously, the matrix $\mathbf{H_{x,v}}$ equals its conjugate transpose, thus it is a Hermitian matrix, satisfying the positive definite requirement of kernel matrix (Gram matrix) in SVM[24,25]. For Iris (Eqs. (9)–(16)) and the nonlinear datasets (Eqs. (17)–(24)), this matrix can also be derived in a similar way, while for the latter the off-diagonal matrix element has a form of $(1 + e^{i(x-v)}) \cdot (1 + e^{i(x+\theta)}) \cdot (1 - e^{-i(v+\theta)})/8 + (1 - e^{i(x+\theta)}) \cdot (1 - e^{-i(v+\theta)})/4$ ($=1$ when the element $x = v$), omitting the coefficients for simpleness. Then, the classification can also be done by using this matrix **H** to solve an optimization problem in the Eq. (33) where the matrix $\mathbf{A'}$ and vector $\mathbf{b'}$ are optimization parameters. Therefore, training this photonic chip can produce an implicit equation related to the kernel matrix.

$$\mathbf{H_{x,v}} = \langle \mathbf{G(x)}, \mathbf{G(v)}^* \rangle = \left[\sum_j\left(|w_j'|^2 e^{i(x_j - v_j)}\right)\right], \sum|w_j'|^2 = 1, j = 0, 1, …, p \quad (28)$$

$$G_0(x_1)G_0(v_1)^* + G_1(x_1)G_1(v_1)^* = \frac{1}{2}\left(e^{i\theta_{01}} - 1\right) \cdot \frac{1}{2}i\left(e^{i(x_1+\theta_{11})} + 1\right) \\ \cdot \frac{1}{2}\left(e^{i\theta_{01}} - 1\right)^* \cdot \left[\frac{1}{2}i\left(e^{i(v_1+\theta_{11})} + 1\right)\right]^* \quad (29)$$

$$G_2(x_1)G_2(v_1)^* + G_3(x_1)G_3(v_1)^* = \frac{1}{2}\left(e^{i\theta_{01}} - 1\right) \cdot \frac{1}{2}\left(1 - e^{i(x_1+\theta_{11})}\right) \\ \cdot \frac{1}{2}\left(e^{i\theta_{01}} - 1\right)^* \cdot \left[\frac{1}{2}\left(1 - e^{i(v_1+\theta_{11})}\right)\right]^* \quad (30)$$

$$\sum_{m=0}^{3} G_m(x_1)G_m(v_1) = \frac{1}{2}\left(e^{i\theta_{01}} - 1\right) \cdot \frac{1}{2}\left(e^{i\theta_{01}} - 1\right)^* \\ \cdot \frac{1}{2}\left(1 + e^{i(x_1-v_1)}\right) \rightarrow |w_1'|^2 e^{i(x_1-v_1)} + |w_1'|^2 \quad (31)$$

$$\sum_{m=4}^{7} G_m(x_2)G_m(v_2) = \frac{1}{2}i\left(e^{i\theta_{01}} + 1\right) \cdot \left[\frac{1}{2}i\left(e^{i\theta_{01}} + 1\right)\right]^* \\ \cdot \frac{1}{2}\left(1 + e^{i(x_2-v_2)}\right) \rightarrow |w_2'|^2 e^{i(x_2-v_2)} + |w_2'|^2 \quad (32)$$

$$\mathscr{L} = \min\sum\left(\mathbf{y_t} - |\mathbf{A'H} + \mathbf{b'}|^2\right)^2/N \quad (33)$$

The difference between ONN and PPC is summarized in Table 1. As seen, the PPC differs from ONN in equation expression, data input, and device operators. It does not utilize activation functions as used by ONN. For PPC, the data to be

**Table 1 Model comparison between ONN and PPC in this work.**

| Model | | ONN | PPC |
|---|---|---|---|
| Algorithm | | ANN | SVM |
| Equation | | $\mathbf{y} = f(\dots f(\mathbf{W}f(\mathbf{W}\mathbf{x})))$[1] | $\mathbf{y} = \mathbf{W}\mathbf{G}(\mathbf{x})^2$ |
| Data (**x**) input | Domain | Optical | Electrical |
| | Coded to | Amplitude or Intensity | Phase |
| | After coding? | $\mathbf{x}$ | $\mathbf{G}(\mathbf{x})$ |
| | Trained? | No | Yes |
| Device operator | | Linear operator $\mathbf{W}$ (matrix) | Nonlinear projection operator $\mathbf{G}$ and linear operator $\mathbf{W}$ |
| Activation function $f$ | | w/ | w/o |
| Loss function | | same | |
| References | | [6–8,14,15,17] | This work |
| Optical bias port for coded data? | | w/ [14,15] | w/o |

[1]$\mathbf{x}$ and $\mathbf{y}$ are the input and output vectors, respectively. $\mathbf{W}$ is the linear operator in matrix format. $f$ is the activation function.
[2]$\mathbf{G}(\mathbf{x})$ is the projection operator in vector format.

analyzed remains in electrical domain and corresponds to a nonlinear mapping function in optical domain after coding, while it is coded into optical domain to denote itself for ONN. For on-chip training, the loss function is not specialized and can have various choices same as ONN if the optical power is used. In addition, for ONN, all-zero data that are coded to the amplitudes with identical phases sometimes need an additional optical bias[14,15], while the PPC has one global optical input. Here we comment on the advantage associated with implementing this projection-based classification principle. This PPC does not require optical nonlinear activation functions. Large-scale silicon photonic circuits can perform VMM very efficiently, but a complete ONN also requires on-chip activation functions that is one of the main challenges in silicon photonics. Heterogenous integration is a possible solution, but optical nonlinear devices usually require high threshold optical powers or consume high power for amplification, consequently scarifying the energy efficiency associated with utilizing photonics. Although several all-optical neurons were proposed[30], so far, photonic ANNs mostly adopted inter-layer optical-electrical-optical (OEO) conversion scheme in both experiment[6,13,14] and simulation[15,17,18,31]. This projection-based idea can be realized only by passive silicon photonic circuits, without adding cost on heterogenous integration or intermediate OEO circuits.

We instantiate the XOR example for understanding above explanation in Supplementary Section 1.1 in which the PPC and ANN are compared. Examples of ANN that can or cannot separate XOR are given in Supplementary Figs. 1(a–c). For simpleness, we directly map the two-dimension bit vector of XOR to a 4-dimensional complex vector by using a $2 \times 2$ MZI as the data input element, as shown in Supplementary Figs. 1(d) and 1(e). Afterwards, a linear matrix transformation in complex domain completes classification and the squared modulus (corresponding to optical power) indicates the label in one-hot vectors. Even though at the final stage detecting the optical power involves in a function of squared modulus, as we explained above, the classification is ready done by a hyperplane in a high dimensional complex space and training the optical power is equivalent to maximizing the distance from the projected vector to the plane in complex space. Training only a linear transformation in complex space is faster than training ANN as

shown in Supplementary Fig. 1(f). This projection-enabled classification can be explained by the SVM-like principle and kernel method and be understood by the examples in Supplementary Sections 1.2 and 1.3.

The MZI-mesh-based (Clements' topology[28]) interferometer circuit performs matrix transformation for the $n$-dimensional projected vector. For a $n \times m$ (if $m < n$) matrix transformation, when all output ports are used in training, we impose an additional condition of energy conservation to this matrix transformation. This condition is satisfied in unitary transformation, but this forcible constraint is not always guaranteed for an arbitrary matrix transformation. Training the matrix using all ports is using a $n \times n$ unitary transformation to approximate a $n \times m$ arbitrary matrix transformation. Therefore, drop out (dropping $m$-ports out) is necessary for correct matrix transformation. Drop-out is usually adopted in convolutional ANN programs to suppress over-fitting and improve generalization capability.

**XOR separation experiment.** The PPC chip using the topology in Fig. 1a was fabricated for proof-of-concept demonstration. Figure 2 shows the fabricated chip with fiber arrays coupled, measurement setup, and packaged module (see Methods). Based on the setup in Fig. 2, we perform on-chip training experiment and compare the BFO and the FP algorithm using RMSprop (root mean square propagation[32]) optimizers (see Methods). We experimentally demonstrate XOR separation, which is a classic example of linearly inseparable problem (see Supplementary Section 1.1), usually being used as a benchmark of nonlinear classification[14,15]. In this work, the data $x$ to be classified is normalized to the phase in unit of $\pi$ and then converted to the voltage parameter by interpolating the power $xP_\pi$ ($P_\pi$ is the $\pi$-shift power) to the measured power-voltage curve (see Supplementary Section 2). For XOR, the bit 0 is set as an off-state voltage ($V_{off}$) that is included as a learning parameter and the bit 1 is converted to an on-state voltage $V_{on}$ by interpolating $P_\pi$ from the power-voltage curve (Supplementary Fig. 3), which is equivalent to using $RV_{on}^2 = P_\pi + RV_{off}^2$ (for thermo-optic phase shifter) where the measured $P_\pi$ (corresponding to the $\pi$-shift voltage $V_\pi = (RP_\pi)^{1/2}$) and resistance $R$ are given in Supplementary Fig. 4. Then, a logic operation $X = B(x_1, x_2)$ ($x_i = 0$ or 1, $X$ is the logic value, B denotes a Boolean logic) becomes to $X = B(V_{off}, V_{on})$ and its voltage parameters are input to two MZIs at the layer $L = 1$ (Fig. 1a) as shown in Fig. 2a. Any two ports can be assigned to represent the logic value $X$. Here we use the ports 1 and 5 to present 0 and 1, respectively. At beginning, all voltages are initialized randomly and then the training is done by either BFO or RMSprop. The BFO training result is shown in Fig. 3a. At start, the power map is random, showing no information of XOR logic, however after training, the power map exhibits a XOR-like pattern. The maximum powers occur at the port 1 for the bit patterns 00 and 11 and at the port 5 for 01 and 10. The port 5 (standing for 1) equals XOR logic values, while the port 1 equals the bar of XOR. RMSprop training shows a similar XOR-like power map in Fig. 3b. This demonstrates that both BFO and RMSprop algorithms are applicable for on-chip training photonic devices. Under current experimental conditions (see Methods), BFO and RMSprop have a similar convergence behavior of MSE errors (shown in Fig. 3c) in the early stage, but the final MSE of RMSprop is larger than that of BFO. As seen in another BFO training experiment in Supplementary Fig. 5(a) that shows a better convergence with even a larger step, BFO can succeed in training under various parameter conditions, showing higher success rate and robustness to experimental conditions than RMSprop, even though the convergence speed could be slightly

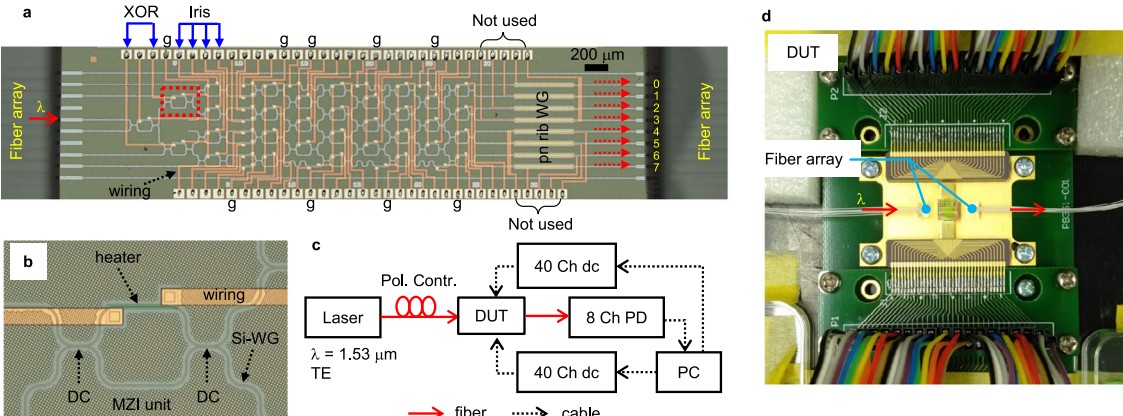

**Fig. 2 Fabricated PPC chip and measurement setup. a** Microscope picture of the fabricated chip with fiber arrays coupled. **b** An enlarged MZI unit in the red square in (**a**). **c** Setup of measurement system. **d** Wire-bonding packaged module. XOR: exclusive OR. The letter g marks out the ground pads. WG: waveguide. The pn rib WG denotes the rib waveguides embedded with pn junctions and their corresponding pads are not used in this study. DC: directional coupler. λ denotes the laser wavelength. TE: transverse-electric polarization. Pol. Contr.: polarization controller. DUT: device under test. Ch: channel. dc: direct-current source. PD: photodetector. PC: computer. Red arrows indicate the light input and output. Blue arrows indicate the data input from electrical domain.

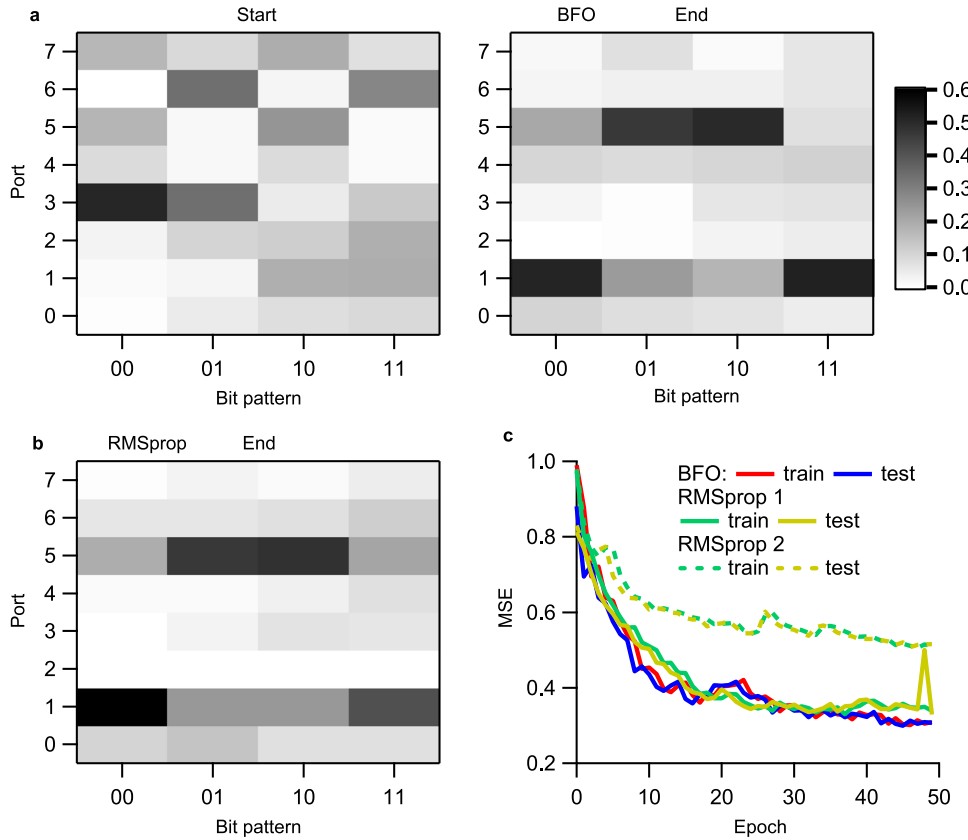

**Fig. 3 XOR (exclusive OR) separation experiment. a** Measured optical power map $\mathbf{y} = \{p_i/\Sigma p_i\}$ (normalized output power) before and after BFO training. $p_i$ denotes the optical power at the port $i$. BFO: bacterial foraging optimization. **b** Measured power map after RMSprop training. RMSprop: root mean square propagation optimizer. All maps have the same scale. **c** Experimental MSE error versus the training epoch for both BFO and RMSprop algorithms. RMSprop cases 1 and 2 use $\Delta V = 0.088$ V and 0.044 V, respectively. The train and test both use the same 2-bit patterns (samples), and the test is verified after every-epoch training. Source data are provided as a Source Data file.

different at the initial stage. When successful, RMSprop has similar training results as BFO, however it sometimes fails under different conditions. For example, if we decrease the voltage step to the half for gradient measurement, RMSprop will fail to complete training within 50 epochs (as seen in Fig. 3c), showing sensitivity to gradient evaluation condition. The learning rate of RMSprop has been optimized in our experiment. For BFO training, the comparison between experiment and simulation on MSE and optical power evolution for each bit pattern are given in Supplementary Sections 3.1 and 3.2. Supplementary Fig. 7 shows the power distribution inside the device to visualize the optical propagation for XOR separation progress.

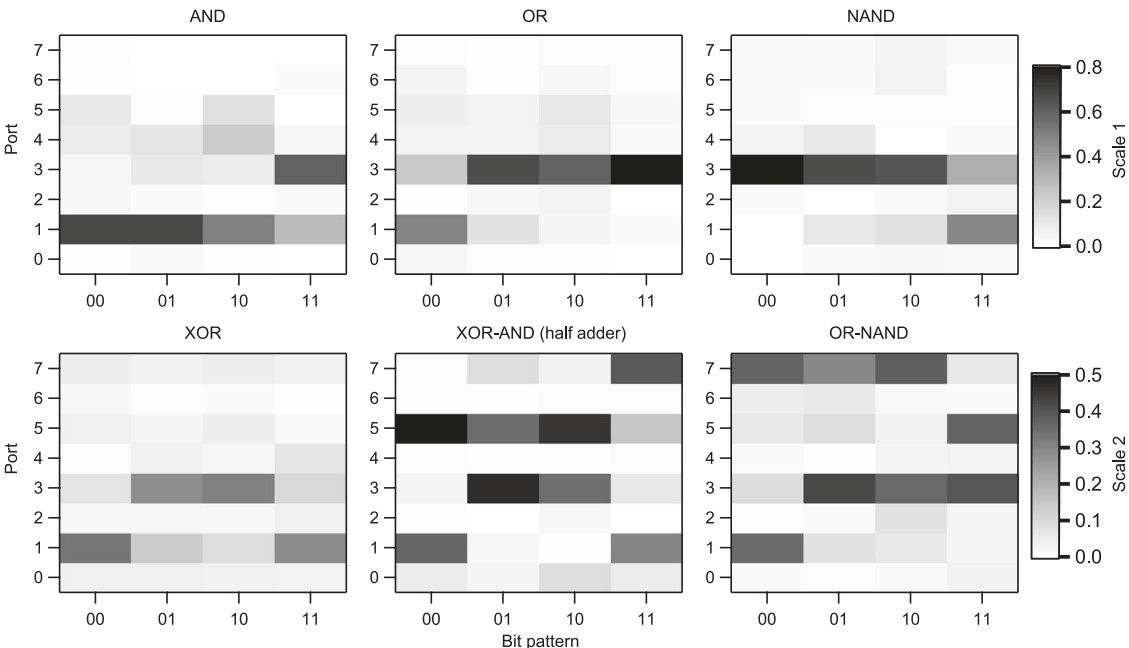

**Fig. 4 Single and combinational Boolean logic operation experiment.** Power maps of single logic (AND, OR, NAND, XOR) and combinational logics (XOR-AND, OR-NAND). All figures are obtained by BFO training. The ports 1 (and 5) and 3 (and 7) denote the logic values 0 and 1, respectively. Scale 1: single logic. Scale 2: combinational logic. XOR: exclusive OR. NAND: not AND. Source data are provided as a Source Data file.

**Other single and combinational logics**. By reconfiguring the voltage weights, this PPC device can also perform other Boolean logics including AND, OR, NAND (not AND). More importantly, this single device can perform combinational logics such as XOR and AND, OR and NAND, simultaneously. For single logics, as seen in Fig. 4, we use the ports 1 and 3 to present the value $X = 0$ and 1, respectively, and the power map can be on-chip reconfigured to either AND, OR, NAND, or XOR logic pattern. For combination logics, we assign two ports to present each logic value (e.g., the ports 1 ($X = 0$) and 3 ($X = 1$) for one logic, the ports 5 ($X = 0$) and 7 ($X = 1$) for the other). As shown in Fig. 4, the multiple logics such as XOR and AND, OR and NAND, can be achieved simultaneously. The XOR-AND logic is a half-adder, which has potential for optical logic computing[33]. In terms of practical use, the differential detecting mode between two ports (presenting $X = 0$ and 1) can be adopted. The experimental results in Fig. 4 well match the simulated results in Supplementary Section 3.3. The port assignment is not unique (as shown in Supplementary Fig. 8) and successful training is achievable in experiment for different port assignment as seen from the XOR patterns in Figs. 3 and 4. In above experiment, all ports are used in calculating the training loss. Dropping out some ports can give better bit contrast as shown in Supplementary Section 3.4. The simultaneous realization of multiple Boolean logics is demonstrated in a single photonic device, which may find applications for realizing advanced optical computing circuits.

**Iris classification experiment**. Next, we perform the Iris classification experiment in the PPC device. The well-known Iris dataset[29] has four parameters and three labels: 0 = Setosa, 1 = Versicolor, and 2 = Virginica, each of them having 50 samples. The data is normalized to $2\pi$ as $x = 2\pi(x - x_{\min})/(x_{\max} - x_{\min})$ where $x_{\max}$ and $x_{\min}$ are the maximum and minimum values in the dataset, respectively. Then it is converted to the voltage by the same interpolation method ($RV(x)^2 = xP_\pi +$

$RV_{off}^2$) as XOR (Supplementary Section 2). The four voltage parameters of $V(x)$ are input to the pads (indicated in Fig. 2a) corresponding to the layer 2 in Fig. 1a. The ports 1, 3, and 5 are assigned to represent the labels 0, 1, 2, respectively. Other training parameters are same as the above XOR experiment (see Methods). Both BFO and RMSprop give similar accuracies as seen in Fig. 5a, while the MSE of BFO training is slightly smaller than that of RMSprop (Fig. 5b). This experimental MSE difference is well reproduced in the simulated MSE in Fig. 5c. Both train and test sets show similar convergence features of accuracy and MSE, indicating no obvious over-fitting in the training. For each algorithm, the measured accuracy and MSE are in good consistence with the simulated ones (Supplementary Section 4.1). BFO and RMSprop are also compared in simulation for all above tasks (XOR, Iris, Iris with drop out) in Supplementary Section 7. BFO and RMSprop show similarity in the whole profile of final learned voltage weights (i.e., phase distribution) (see Supplementary Section 4.2). Their discrepancy at some heaters may be related to the relatively large voltage steps (~0.088 V for RMSprop) used in our current experiments. The power map obtained by BFO training is shown in Fig. 5d in which the maximum-power ports are clearly classified into three groups, corresponding to the three species of Iris. The training and verification accuracies are 94.44% and 96.67%, respectively. If dropping three ports out as the output vector, we confirm that up to 98.89% for training and 98.33% for verification (~98.67% for all samples) can be achieved in experiment (Supplementary Fig. 15). These accuracies are comparable to that (97.3%) obtained by complex photonic neural networks[14]. For BFO, the final voltage steps are about 0.123 and 0.097 V for without and with drop out, which indicates that this algorithm does not require very fine voltage steps in current Iris experiment even though they may influence the roughness of converged MSE. For comparison, a $4 \times 5 \times 3$ ANN using rectified linear unit (ReLU) activation functions and a softmax output layer is prepared to classify the Iris dataset using the same train and test sets. This ANN shows a maximum verification accuracy of 96.67% (see

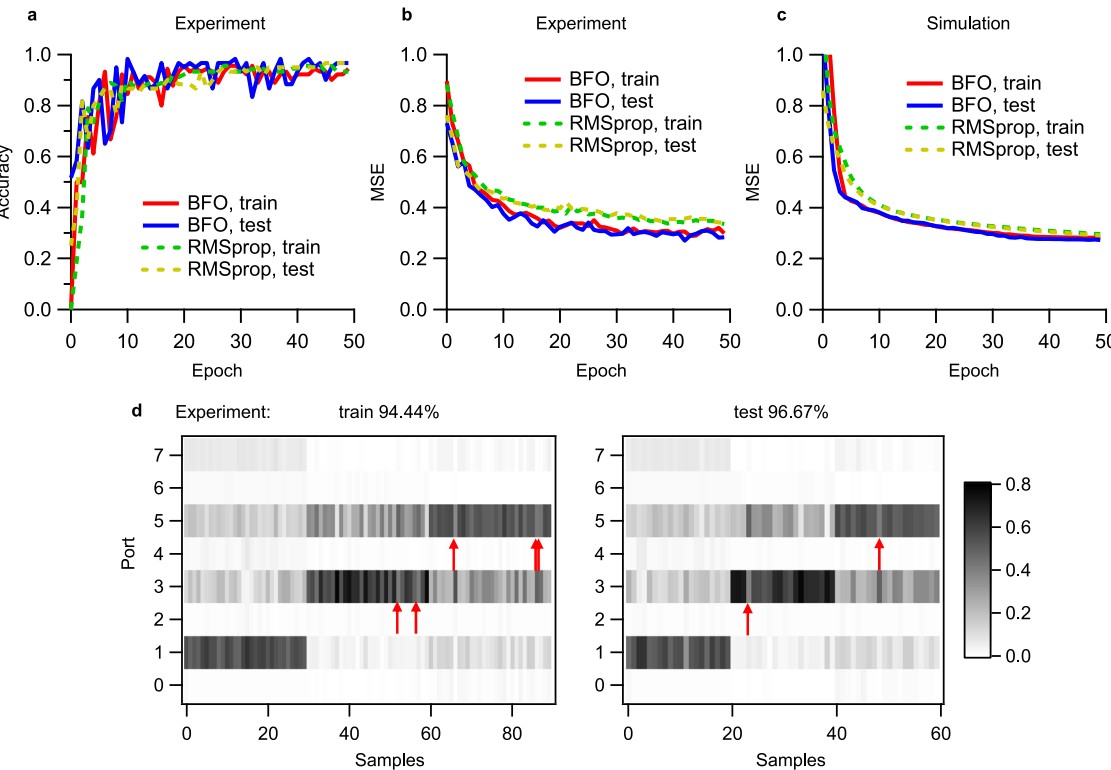

**Fig. 5 Iris classification experiment.** Comparisons between BFO and RMSprop training: **a** Experimental accuracy; **b** Experimental MSE; **c** Simulated MSE. BFO: bacterial foraging optimization. RMSprop: root mean square propagation optimizer. **d** Measured power map (normalized output power $\mathbf{y} = \{p_i/\Sigma p_i\}$, $p_i$ denotes the optical power at the port $i$) of the train set and test set after BFO training. Red arrows indicate the wrongly recognized samples. The train and test sets are explained in Methods. Source data are provided as a Source Data file.

Supplementary Section 4.3), which is comparable to the experimental results obtained by the PPC device.

**Comparison with ANN for nonlinear dataset classification.** We compare the PPC and ANN by simulation for classifying the nonlinear datasets of Circle, Moon, and Spiral. There are two inputs ($x_1$, $x_2$) for these datasets, two classes for the Circle and Moon, three classes for the Spiral. ANN models are explained in the caption of Fig. 6. The PPC implements the projection scheme in Fig. 1d by inputting two variables into cascaded MZI (input $x_1$ to the 1st MZI at the layers 1 and 2, input $x_2$ to the 2nd MZI at the layer 1 and the 3rd MZI at the layer 2, the layer number is shown in Fig. 1a). The mapping function $\mathbf{G(x)}$ is given in the Eqs. (17)–(24). In this simulation, VMM adopts an 8-layer structure of MZI mesh (column × row = $4 \times 4 + 4 \times 3$) and does not use external phase shifter for all MZIs. Thus, the PPC has 43 parameters in total. Both ANN and PPC are trained by RMSprop with the same learning rates for comparison. As shown in Fig. 6a, the ANN with 10 middle nonlinear nodes can classify the Circle but predicts a hexagon boundary and cannot classify the Moon and Spiral. Both the ANN with 50 middle nodes (Fig. 6b) and PPC (Fig. 6c) get 100% accuracy for the Circle and Moon and ~90% accuracy for the 3-classes Spiral. For the Moon and Spiral, the PPC shows obviously different nonlinear boundary features from the ANN. As seen in Fig. 6d, the PPC has almost same convergence speed as the ANN for the Circle, while converges much faster than the latter for the Moon and Spiral. The PPC (trained by FP) can achieve comparable accuracies to the ANN (trained by BP) even with 43 parameters, only ~1/6–1/7 scale of the ANN, and without nonlinear activation functions, indicating scalability advantages.

## Discussion

First, we discuss speed, training time, and power consumption. The classification response time of PPC includes the time of data input, on-chip optical delay, and result readout. In our fabricated chip, optical delay is about 30 picoseconds. This latency from signal input to result readout is much faster than transistor-based electronic hardware and remains same no matter for logic calculation or complex classification computing. The time of the data input and readout is the main restricting factor in our current setup. For the data input, the time is mainly limited by our apparatus of 40-channel current sources that require 2–3 milliseconds (ms) for setting one channel (one heater). For data readout, the time is limited by USB (universal serial bus) communication. This study is a proof-of-concept demonstration and thus we use thermo-optic MZIs for data input, but for practical use, the device should adopt high-speed modulators and integrated photodetectors for data input and readout, respectively. An ideal form should be with high-speed modulators and photodetectors integrated on chip and with drivers and transimpedance amplifiers packaged. Currently, >25 GHz bandwidth can be achieved for both modulators and detectors on standard silicon photonic platforms; thus, the data can be processed at a speed much faster than traditional CPU (central processing unit). With such an integrated form, the training algorithms can be run standalone in digital circuits such as FPGA (field programmable gate array) and DSP (digital signal processor), instead of computer.

The current training time is also limited by the current setting time (2–3 ms for one channel) of our current sources, limiting on-chip training for a large dataset, which can also be improved if integrating drivers for heaters[34,35]. If the driver has a >100 MHz bandwidth, the training time will be dominated by the

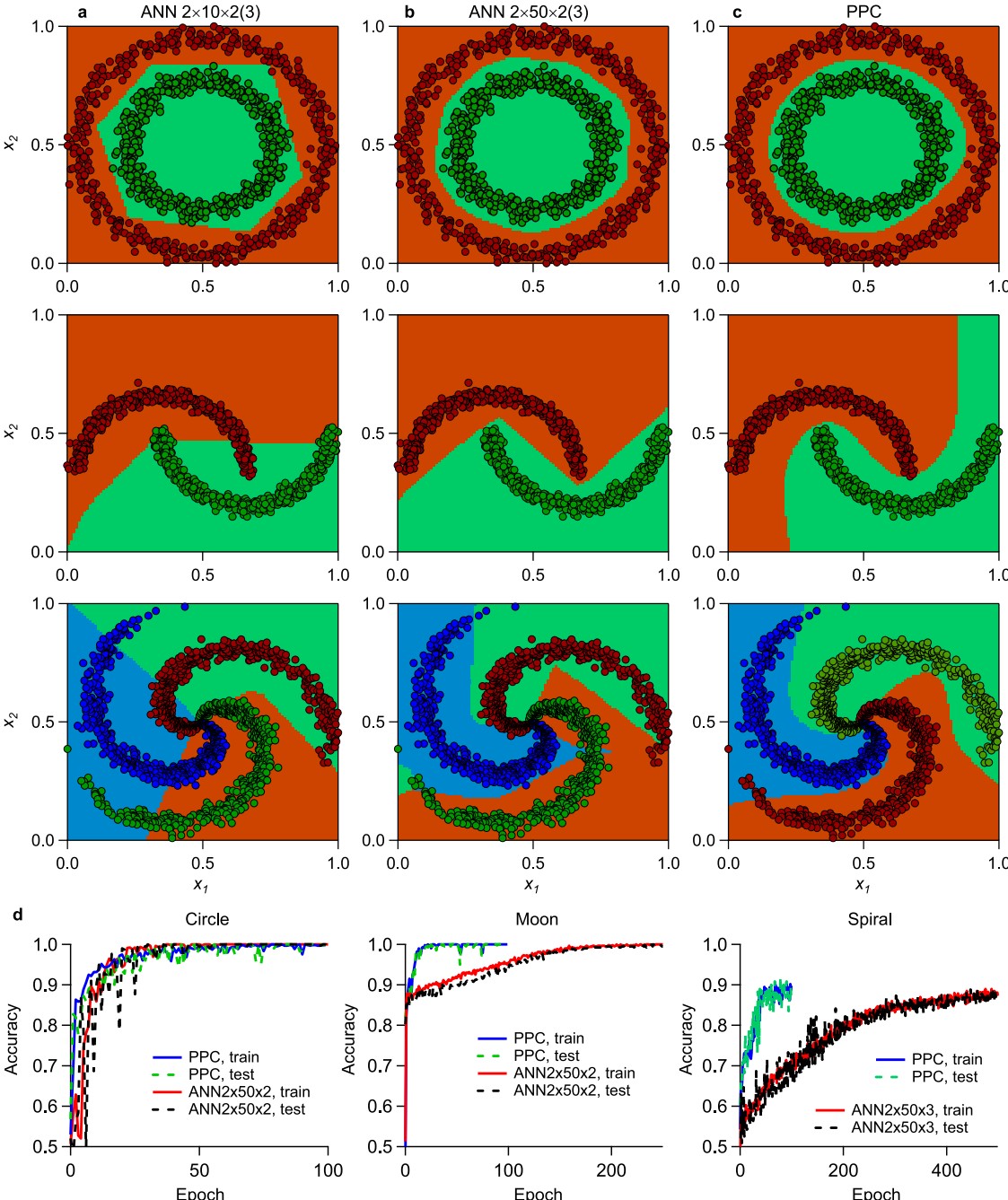

**Fig. 6 Comparison between ANN (artificial neural networks in form of multilayer perceptron) and PPC (projection-based photonic classifier) for classifying the nonlinear datasets: Circle, Moon, and Spiral.** Predicted nonlinear boundary for the Circle, Moon, and Spiral datasets in each column of subfigures: **a** ANN with 10 middle nodes; **b** ANN with 50 middle nodes; **c** PPC. **d** Comparison on the training curves of accuracy between the PPC and ANN with 50 middle nodes. The region of $x_1, x_2 \in [0,1)$ is discretized in a step of 0.01. The data are normalized to $\pi$ for the PPC. In (**a–c**), the colors orange, green, and blue in data points and regions represent the classes 0, 1, 2, respectively. The ANN adopts three-layer models: input (2 nodes), middle (10 or 50 nonlinear neuron nodes), and output (2 nodes for the Circle and Moon, 3 nodes for the Spiral), denoted by ANN $2 \times 10 \times 2(3)$. The PPC has 43 parameters in total and the ports 1, 3, 5 are dropped out as the output vector to correspond to the three classes 0, 1, 2. The ANN has 252 and 303 parameters for $2 \times 50 \times 2$ and $2 \times 50 \times 3$, respectively, and uses ReLU (rectified linear unit) activation functions for the middle layer and softmax output layer. For the Moon and Circle, 1000 samples in total, 600 for training, 400 for testing; for the Spiral, 1500 samples in total, 900 for training, 600 for testing. All training and testing data plotted in (**a–c**), are uploaded and both training and testing accuracies are shown in (**d**). Source data are provided as a Source Data file.

heater response, which is about tens of microseconds (μs) (~30 μs for our heaters, which can be shortened to <10 μs[34]). Assuming 50 μs for one loop, the time will be ~5 s for $10^5$ loops, which is nearly ten-thousand times faster than the current on-chip training. In our current experiment using BFO (in Fig. 3a and Fig. 5), 50 epochs have 3842 and 3690 loops in total for XOR and Iris

classification, respectively (each loop means one time that the device state is changed). Phase shifters with faster response such as carrier-injection[36,37] and phase changed material (PCM) types[8,38] could be adopted to further reduce the training time.

The power consumption mainly comes from the thermal power consumed by the heaters for maintaining the learned

optical phases. As seen in Supplementary Section 4.2, we calculate the power from the voltage distribution learned by BFO and RMSprop, respectively. The total powers are of ~364.5 and 358.7 mW for BFO and RMSprop, respectively. With such a sub-Watt power, the PPC device can be reconfigured to perform logic computing and complex classification that usually needs modeling ANN in electronic processors. An advanced CPU (e.g., Intel Core i7 series) has a typical power at least higher than 30 W; thus, the PPC device offers high energy efficiency for specific machine learning applications. The power scalability is investigated by MNIST simulation in later discussion on the scalability and in Supplementary Section 6.

Second, we discuss the robustness to imperfect control, stability, and reproducibility. This device works on optical phase control; thus, the phase deviation from the learned phase conditions will influence the classification accuracy. The experimental robustness analysis is done for the solutions of voltage weights (in Supplementary Fig. 11) obtained by both BFO and RMSprop algorithms. We intentionally introduced random bias errors to the learned voltage weights and then re-loaded them into the device to measure the accuracy and MSE for both the train and test sets of Iris dataset (see Supplementary Section 4.4). The measured accuracy and MSE in relation of the bias error are shown in Supplementary Fig. 13. As seen, the accuracy remains almost unchanged for <3% bias errors, however, an obvious degradation is observed for >3% bias errors, for both BFO and RMSprop. For the bias error <7%, the solution of BFO is more robust than that of RMSprop because BFO has a slower decreasing speed in accuracy. Therefore, <3% bias control precision is required to avoid accuracy degradation. For classification with drop out, the accuracy can be enhanced by 1–2%, while the bias error allowance will decrease. As seen in Supplementary Fig. 15, an obvious decrease in accuracy is observed for >1% bias deviation.

For stability, the learned voltage weights for Fig. 5 were repeatedly loaded into the device for more than 1 week to check the accuracy and MSE. Within 1 week, both the accuracy and MSE have no obvious degradation (see Supplementary Section 5.1). The variations in accuracy (±1%) and MSE (±0.02) may be related to small polarization fluctuations in single mode fibers. Furthermore, for the stability of each individual port, we automatically reconfigured the device using the BFO algorithm to route the light to each individual port and measured the residual MSE within 1 week. As seen in Supplementary Section 5.2, the MSE almost keeps constant for each port. Above experiment manifests the long-time stability and reproducibility of measurement results.

We confirmed two PPC chips capable to reproduce above experimental results. All experiments were performed in normal lab environment without any thermal management (Supplementary Section 5.3). One reason for the chip reproducibility is due to our advanced 12 inch silicon photonic platform (AIST-SCR) on which our large-scale photonic switches were fabricated with high reproducibility[34,35]. The other reason is that the PPC device for classification application presented in this work inherently has higher robustness to device imperfections than other devices for switching applications (see Supplementary Section 5.4). In silicon photonics, the directional coupler (DC) is one of the most sensitive components and usually deviates from an ideal 3 dB due to fabrication error, inducing crosstalk for optical switches. We introduce a DC deviation $\delta$ as seen in the equation in Supplementary Fig. 19 and then perform training for XOR and Iris tasks. Even with an error up to $\delta = 10\%$, the final MSE is almost same as that without error for XOR, and even lower for Iris (in Supplementary Fig. 19(c)). Almost same classification results can be obtained despite of DC errors, indicating

an inherent robustness to fabrication error, which contributes to reproducibility of the PPC device.

Finally, we quantitatively study the scalability of accuracy and power consumption for the projection-based approach by MNIST (handwritten digit dataset[39]) simulation in Supplementary Section 6. We perform MNIST classification using four architectures (in Supplementary Fig. 20) with gradually increasing the total parameters (i.e., PS). For the architectures of quadratic mapping, a testing accuracy of ~90–91% is achieved with 815 parameters and 16 input components, which benchmarks a >90% accuracy with only <900 parameters, and a ~94% testing accuracy is achieved with 1663 parameters and 32 input components, as shown in Supplementary Fig. 21(a). It is known that such accuracies cannot be achieved by ANN without nonlinear activation functions[14,17], which evidences the role of projection. Implementing nonlinear projection can contribute to ~4–5% accuracy enhancement for MNIST classification. These accuracies are comparable or slightly higher to/than those in previous works: ~93% for 2-layers ANN[17], 90.5%[16], ~93.4%[12], 90%[7], and lower than 98%[18]. The scalability of accuracy follows a relation of log($N^{0.13}$) for training and log($N^{0.09}$) for testing to the total parameter number $N$. There are minimum MZI layer numbers for VMM to reach the maximum classification accuracy no matter with and without external PS, and further adding more duplicate parameters cannot enhance the maximum accuracy (see Supplementary Sections 6.2 and 6.3). With increasing the scale to improve the classification performance, the power is also increased, which can be calculated from the learned phase distribution after training. The power (i.e., absolute sum of total phases) linearly increases with the total parameter number $N$. As shown in Supplementary Fig. 21(b), if using the same thermo-optic PS (assuming a $\pi$-shift power $P_\pi = 15$ mW), the power will be ~5 W for 1663 parameters, which is still much lower than current GPU (graphics processing unit) or CPU for performing such complex classification tasks. If other PS with high energy efficiency such as pin-type[36,37], PCM-type[8,38], or MEMS-type[40] are used to replace the heaters, the total power can be further decreased. The pin-type phase shifter using carrier-injection usually has a $P_\pi$ about 2–3 mW, only 1/5 of the thermo-optic one; then the total power can be decreased to ~1 W even when the device is scaled up to 1663 parameters. In addition, the PCM and MEMS PS can implement VMM with low loss and powerless standby states. Such scales of architectures are achievable on current silicon photonic platforms and the PPC also favors scalability because it consists of only passive photonic circuits without requiring heterogenous integration.

## Methods

**Device fabrication.** The fabricated device shown in Fig. 2a was fabricated on a 220 nm-thick silicon-on-insulator wafer by standard process in the AIST-SCR CMOS line. The silicon waveguide is a fully etched 430 nm-wide wire waveguide and the clad thicknesses are ~1.5 μm. All PS are thermo-optic, using titanium nitride micro heaters. The MZI is composed of two 3 dB DCs and only upper arms were fabricated with heaters as seen in the enlarged MZI unit in Fig. 2b. The electrode metal is ~1.5 μm thick aluminum-copper alloy. The pads of MZIs for inputting XOR and Iris data are indicated in Fig. 2a. This device has 46 heaters in total, 32 for MZIs, 8 for phase bias, and additional 6 heaters for phase error compensation (see Fig. 1a). The ground pads are marked out by the letter g. Note that the rib waveguides with embedded pn junctions are not used, so their pads can be neglected (marked as not used in Fig. 2a), but they can work as on-chip polarizers for polarization adjustment. The light input port and eight output ports are marked by an arrow and numbers (0–7), respectively. For light coupling, inversed tapers were fabricated at both facets, and the chip was packaged with fiber arrays as seen in Fig. 2a. With a straight reference waveguide, the fiber-to-fiber loss including two facets was measured to be ~4.5 dB.

**On-chip training experiment.** Figure 2c shows the setup for on-chip training experiment. In this work, we use the wire-bonding packaged module in Fig. 2d, which offers more stable electrical contacts than using 40-pin probes[21]. The

computer read the optical powers via an 8-channel photodetector array and ran the training algorithms to control all heaters via two 40-channel direct-current (dc) sources. The laser at the wavelength of 1.53 μm was input to the chip after tuned to TE polarization. The experiment was done at room temperature without special thermal management. Supplementary Section 2 explains the details for controlling the dc sources and preparing the input data. We compare the BFO with RMSprop for on-chip training. BFO is more implementable than RMSprop in experiment because the latter requires changing the parameter by a small enough step for accurate gradient evaluation, while in experiment a small step is more susceptible to noises, making accurate gradient evaluation again difficult and thus sometimes resulting in failure of training. The BFO training is stochastic in nature and not so strict on the parameter step. Our results show that the BFO has a higher success rate than RMSprop in on-chip training experiment and its training results are more robust to imperfect control than the latter. The source code of BFO is shown in Supplementary Section 9, for which we use 10 bacteria, 5 chemotaxis loops, and 20 maximum swimming steps in experiment. A bacterium means a state vector, and the bacteria keep alive in chemotaxis loops, while they will be reproduced once the chemotaxis loop is finished. Swimming means the bacteria state update along an error-decrease direction until the error increases[22]. We used an adaptive voltage step $\Delta V = 0.044 + 0.264\delta$ ($\delta$ is the MSE loss at each training epoch). For RMSprop, the gradient for each heater was evaluated using $\Delta V = 0.088$ V or 0.044 V (for comparison) and then the weights (i.e., voltage distribution among all heaters) were updated using the well-known RMSprop procedure[32]. A training epoch means the training dataset has been completed through the entire algorithm structure, for which BFO runs over 10 bacteria, 5 chemotaxis loops, and some swimming steps (<20), and RMSprop completes gradient measurement by iterating all heaters. The learning rate of RMSprop was optimized in experiment (set to 30 since the DAC is the learning parameter as seen in Supplementary Section 2). The voltage was randomly initialized within the voltage region shown in Supplementary Section 2 for all experiments. The control program was built on Python and each algorithm called the same device module that was encapsulated as a class based on PyTorch[41]. For visualization, Supplementary Videos 1, 2, and 3 show the BFO optimization process of searching the minimum of a 10-dimensional sphere function, XOR separation experiment, and automatic port configuration experiment, respectively. As for the Iris dataset, the train set contains the samples of number 0–29 (Setosa), 50–79 (Versicolor), and 100–129 (Virginica), and the test set uses the left numbers 30–49 (Setosa), 80–99 (Versicolor), and 130–149 (Virginica).

**On-chip training simulation**. A common device model was also built on Python using PyTorch[41] and both algorithms called the same model. This model simulates the device as a black box. The algorithm implementation is exactly same for both experiment and simulation. The only difference is that in simulation, the algorithm takes the phases as the learning parameters, but in experiment, it takes the voltages as the learning parameters. For BFO, an adaptive phase step $\Delta\phi = 0.03\pi\delta$ was adopted ($\delta$ is the MSE loss). For RMSprop, the gradient was evaluated using $\Delta\phi = 0.0001\pi$ and the learning rate was 0.005. The phase was randomly initialized within $\pm0.3\pi$ for all simulations.

## Data availability

All data generated in this study are provided in the Source Data file. Source data are provided with this paper.

## Code availability

The code that supports the findings of this study are available from the corresponding author upon request.

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

## Acknowledgements
This work was supported by Japan Science and Technology Agency (JST), CREST Grant Number JPMJCR15N4 and JPMJCR21C3, Japan. The authors thank all staffs at the AIST-SCR station for device fabrication.

## Author contributions
G.C. proposed the concept, designed the chip, and developed the code for experiment and simulation. G.C., N.Y., and K.Y. built the experimental setup and performed the experiment. Y.M. and M.O. assisted with process and fabrication. G.C. and S.K. assisted with packaging. T.I. and S.N. commented on the principle and data. This paper was written by G.C. with contributions from all other authors.

## Competing interests
The authors declare no competing interests.
