## [Peer Review File · Nature Communications]

On-chip bacterial foraging training in silicon photonic circuits for projection-enabled nonlinear classificationREVIEWER COMMENTS

Reviewer #1 (Remarks to the Author):

The paper titled "Performing classification in silicon photonic circuits by implementing nonlinear projection" presents a silicon photonic implementation of a classifier, which is trained with the help of an external, classical electronic computer, to read the optical outputs and to generate the new weights for performing the inference in optics.

Two key ideas are presented: resort to a non-linear mapping, precomputed and pre-existing, and a passive linear matrix-vector analog computation, based on Mach-Zehnder Interferometers.

The idea is convincing and well motivated. It is similar, in concept, to that used in photonic implementation of Reservoir Computing, a more mature application.

I believe that the both the new photonic hardware and the associated results obtained on XOR separation and Iris data classification are convincing enough, even if the considered classification problems are among the most entry-level ones in the machine learning community. As admitted by the authors, the work is a small-scale, proof-of-principle demonstration and nothing more, although interesting.

I also found the "supplementary information" really helpful in better understanding the implementation details, along with the video for XOR training, which is really interesting. Thus I recommend for the publication.

Below are a few minor comments:

- On page 2, line 40: please replace "principle" with "principal" (component analysis)
- On page 7, line168: please add "than" after "...FPGA is usually larger" and "1 W at the ..."

Concerning the supplementary material, the minor changes are:

- On page 3, revise the English in the sentence "Based on only passive photonic circuits can this device be realized."

Reviewer #2 (Remarks to the Author):

The authors claim a new nonlinear projection operator implemented by phase-encoded inputs trained on small datasets and using an SVM kernel method. Additionally, the authors experimentally demonstrate training methods such as BFO on such datasets.

High-level comments:

Why SVM?: Some more discussion is needed to compare SVM and ANN at a high level for the physics audience of the paper. Assuming we are building this into a large-scale application, why should we care about implementing an SVM if ANNs can already do the job? Perhaps an argument can be made about training speed at scale, not just for small 2d problems (see following comment). You also need to properly define an ANN and SVM. An SVM is actually in theory a type of ANN (if defined as any nonlinear function with matrix multiplies in it), so it isn't clear if you are doing anything novel in that sense.

Scalability: While I understand that the datasets that can be tested on this chip are always small, at least one simulation of a larger chip should be performed to demonstrate the whether the results of Fig. 5 scale to reasonably size machine learning problems of interest. SVMs are generally known to train faster than neural networks and are parametrized by only a relatively small number of support vectors in the datasets. For example, a similar comparison (multi-layer ANN vs

nonlinear projection) might work on a dataset like MNIST. You might try Fourier-MNIST to keep the dataset to a reasonable size and try to limit the number of parameters and/or layers (see Williamson, et al., 2020 (cited) and Pai et al., 2020 "Parallel programming..." for a two-layer approach). I would say this (or some similar argument) is required for publication, i.e. to complete the argument of scalability in the Discussion section.

Training: The demonstration of BFO training in a photonic chip is interesting and should be emphasized more; I know you have a previous paper on the theory but you provide a very useful demonstration where you assess the effect of phase noise on the training. You might bring some results (e.g. comparison of forward finite difference / RMS prop to BFO) from the Supplement into the Main paper. Some more results and visualization of the BFO process might also be warranted. I think this will help clarify also the novelty of this paper: that a ****trainable**** matrix-vector multiply can be implemented within a photonic network of MZIs. If the authors can find a title and reorganize the paper around that thesis, it would much be easier to follow and appreciate the contribution of the work.

Validity of statistical analysis:

Generally, the machine learning approaches should always have both training and testing/verification dataset curves to account for possible overfitting. For the plots in Fig. 5 in particular, I think this should be done.

Reproducibility:

The experiment appears to be reproducible and in line with other implementations, but I have a question about an experimental detail. Thermal control is typically required to ensure that the heat transfer throughout the chip doesn't introduce any spurious phase shifts throughout the device. It is unclear if this is part of the setup and whether it affects training. The authors should elaborate on this point.

Organization:

The paper lacks some impact structure and therefore struggles to bring home the core message and novelty of the work. Mainly, I have to do a lot of work to figure out whether this work really adds any value above what is already in the field. For example, after the results are discussed, there is some valuable analysis, but there is no discussion to compare those results to known benchmarks in the field (Shen et al. 2017, Zhang et al. 2021). There is also insufficient discussion of the core differences of SVMs and ANNs as it relates to real-world applications; it seems hand-wavy in the Supplement's Scalability section with little data/experiments to back it up.

Specific comments:

Figures:

Fig 5- is a bit too busy, please limit to fewer plots with bigger axes.

Fig 4- colormap should be changed to a non-rainbow colormap for clarity

Fig 3- The colors for each port are a bit distracting. You could make all the bars that are not part of the classification be a gray color and highlight the bars we are supposed to pay attention to with a blue or red color. Alternatively, use a 2d colormap instead.

Fig 1- There is a strange redundancy in your architecture. If you have two MZIs next to each other, you can reparametrize the system as a single MZI. You can just flip the architecture so that doesn't happen. Also in case you wanted to show a universal 8x8 architecture in the figure,

Equations:

Re-define B in Eq 4 for clarity in terms of the terms in Eq 1

I do not fully understand Eq 4, if B is a matrix and H is a scalar inner product, then how does it match the shape of B? There might be a notation issue here.

Other:

The authors mention some alternatives to heaters for phase shifters. I urge reading work on MEMS phase shifters and phase change materials which are both low-loss and take up no power in static configuration. A sentence or two should cover that in the Discussion.

Summary:

Publication of this paper depends on differentiating what is now a common photonic architecture (MZI networks) from existing implementations. The authors attempt to do this by framing the architecture as a nonlinear SVM kernel implementation that can be trained from scratch, but I am not yet convinced that it reaches the necessary level of impact for publication in Nature Communications. We already know these chips can do matrix multiplication, so applying it to an SVM (which is basically just another ANN, or really any machine learning model with some matrix-multiplication) doesn't seem impactful enough on its own, and free space implementations seem to do much better by scaling the SVM to much larger matrix sizes than can be achieved on a chip. Therefore I cannot recommend publication in the current form, but perhaps with significant revision I could be convinced otherwise. The authors should emphasize the more impactful training results, perform more simulations on the SVM architecture (applied to bigger multi-dimensional datasets like MNIST), and reorganize the paper to more clearly lay out the advantages of the overall approach over existing neural network architectures (this should be in the main paper, not supplement).

That said, it is an impressive feat whenever a group manages to implement a matrix-vector product on a photonic chip. Any optics journal would be suitable to publish this work as is if the authors want to stick with SVM/nonlinear projection as the main contribution.

Responses to the reviewers

We have fully considered the reviewers' comments and all reviewers' concerns and questions have been systematically studied with more experimental and simulated works. We have made substantial and thorough revisions to our manuscript and Supplementary (Suppl.) file with adding more comprehensive data, examples, and detailed explanation, as you can see in the revised manuscript and Suppl. file. In this response letter, all questions and comments are addressed below in a reproduced verbatim style (the reviewers' comments are cited in *Italic* format and answered in a point-by-point style). For your quick view, I would like to briefly list the main points, while I would like you to review the details in the manuscript, Suppl. file, and this response letter.

- (1) During this time, we chose another chip and completed wiring bonding. This revised manuscript is based on this wire bonding chip. We repeated all previous experiments with this new chip.
- (2) The novelty and impact of this work have been clearly explained in the new introduction and Suppl. Section 1. We demonstrate a new on-chip training method and propose a new classification principle for photonic circuits to implement machine learning. In most previous demonstrations, photonics only performed inference and on-chip training remains less demonstrated. We demonstrate on-chip bacterial foraging training in silicon photonic circuits for projection-enabled nonlinear classification. This novel principle can offer comparable performances to ANN for various benchmarks even with much smaller scales and without leveraging nonlinear activation functions, showing scalability advantage. This principle is a more SVM-like principle than ANN, which has not been implemented in photonics so far. (If thinking it were exactly same as ANN, its capability to classify nonlinear datasets even without activation functions would contradict to ANN that cannot give such high performances if without activation functions, manifesting the role of projection and evidencing the novelty of this work). We provide all details of the proposed approach and comprehensive data to evidence its advantages. This work paves a novel way for photonic devices to perform nonlinear classification without leveraging activation functions.
- (3) This novel approach can achieve test accuracies up to ~98.3% for Iris, 100% for Circle, 100% for Moon, >98% for 2-classes Spiral, ~90% for 3-classes Spiral, ~90% for MNIST with 815 parameters, ~94% for MNIST with 1663 parameter, by utilizing only passive circuits. For the first time we experimentally demonstrate combinational logic (XOR-AND, OR-NAND) in a single photonic chip, in addition to all single logics. Benchmark comparison with a previous paper is summarized in Suppl. Section 8 and described at corresponding positions in the main text.
- (4) All concerns on scalability, reproducibility, robustness, and long-time stability have been systematically studied and verified with adding more quantitative data. All data and core codes are uploaded.

To Reviewer 1:

Comment 1:

The paper titled "Performing classification in silicon photonic circuits by implementing nonlinear projection" presents a silicon photonic implementation of a classifier, which is trained with the help of an external, classical electronic computer, to read the optical outputs and to generate the new weights for performing the inference in optics.

Two key ideas are presented: resort to a non-linear mapping, precomputed and pre-existing, and a passive linear matrix-vector

analog computation, based on Mach-Zehnder Interferometers.

The idea is convincing and well motivated. It is similar, in concept, to that used in photonic implementation of Reservoir Computing, a more mature application.

Answer 1: Thank you very much for your kind comments and understanding our idea. To make our work better for easy understanding, the principle was re-explained in details and the manuscript was re-organized. Detailed explanations are provided in Suppl. file. (Reservoir is usually regarded as a recurrent neural network instead of SVM, while it can also be regarded as an implicit projection-based algorithm utilizing time-dynamics induced mapping, which could be distinguished from our proposal as explained in *principle* section in the main text and Suppl. Section 1).

Comment 2

I believe that the both the new photonic hardware and the associated results obtained on XOR separation and Iris data classification are convincing enough, even if the considered classification problems are among the most entry-level ones in the machine learning community. As admitted by the authors, the work is a small-scale, proof-of-principle demonstration and nothing more, although interesting.

I also found the "supplementary information" really helpful in better understanding the implementation details, along with the video for XOR training, which is really interesting.

Thus I recommend for the publication.

Below are a few minor comments:

- On page 2, line 40: please replace "principle" with "principal" (component analysis)

- On page 7, line168: please add "than" after "...FPGA is usually larger" and "1 W at the ..."

Concerning the supplementary material, the minor changes are:

- On page 3, revise the English in the sentence "Based on only passive photonic circuits can this device be realized."

Answer 2: Thank you very much for your interests in our work. I am sorry for these typo error and since the main text and Suppl. file have been substantially re-written, these sentences with errors have been deleted or revised. The line 40 is deleted. The line 168 is now at Page 12, Paragraph 3. The last sentence you mentioned is now in the main text, at Page 4, Paragraph 1, Line 8.

To Reviewer 2:

Comment 1:

Reviewer #2 (Remarks to the Author):

The authors claim a new nonlinear projection operator implemented by phase-encoded inputs trained on small datasets and using an SVM kernel method. Additionally, the authors experimentally demonstrate training methods such as BFO on such datasets.

High-level comments:

Why SVM?: Some more discussion is needed to compare SVM and ANN at a high level for the physics audience of the paper. Assuming we are building this into a large-scale application, why should we care about implementing an SVM if ANNs can already do the job? Perhaps an argument can be made about training speed at scale, not just for small 2d problems (see following comment). You also need to properly define an ANN and SVM. An SVM is actually in theory a type of ANN (if defined as any

nonlinear function with matrix multiplies in it), so it isn't clear if you are doing anything novel in that sense.

Answer 1: Thank you very much for your valuable comments which enlighten us to explain our proposal, novelty, and why we consider implementing it in a better and more understandable way. Here we would like to answer your questions in the following two aspects. We have added the explanations below in the section of Principle and device topology at Pages 5-7, Suppl. Section 1, and introduction at Page 3, Paragraph 2.

(1) Why SVM?

This question includes two aspects: (a) why we explain this photonic device as SVM-like classifier instead of ANNs, and (b) why we should consider implementing it. For nonlinear classification, the idea of SVM is to use a pre-chosen nonlinear function to map the data from the input space into a higher-dimensional feature space to seek an easy linear separation. Fig. A1(a) shows a quadratic mapping function to illustrate this idea. As seen, after projection, the boundary points are projected to form a line (not shown) that linearly separates two classes of data with a margin δ . Similar idea can be implemented by utilizing sinusoidal functions which are well known as phase-amplitude nonlinearity in photonic devices. As shown in Fig. A1(b), using sinusoidal projection, multiple clusters of binary-class data can be easily separated by a line which is similarly determined by boundary points (Support vectors). This sinusoidal mapping function can be used as a quadratic-like one if limiting the phase range. Furthermore, linear combination of sinusoidal functions or their multiplications can be utilized to perform more complex classification. It is this idea using sinusoidal projection (from phase to complex amplitude domain) that is what we implement in our photonic device and that is why we explain it as SVM instead of ANN, because obviously such an effect associated with sinusoidal projection is not utilized in ANNs.

We agree with you about the similarity in mathematical formula between the simplest cases of ANN and SVM, however, this projection-based separation via directly treating the input data by a mapping function is the fundamental idea of SVM and usually is not included in the conventional ANN frame. Even if we use the same nonlinear function for mapping in SVM and for the activation function in ANN with matrix multiplies in it, it will be like $x = f(wx)$ for ANN including a first linear dimension expansion and a latter nonlinear treatment, but it will be like a first mapping $x \rightarrow \{x_1x_1, x_1x_2, x_2x_1, x_2x_2\}$ (*e.g.*, for the case of quadratic function) and a latter $y = wx$ for SVM. Obviously, the mapping operation in the latter cannot be expressed as $f(wx)$ and such a mapping does provide above projection-assisted separation effects. In addition, this mapping function has wide choices corresponding to various kernel functions: polynomial, RBF, *etc.* (see Refs. [A1-A3]). Therefore, the ANN frame cannot fully cover SVM. They are usually regarded as two distinguishable machine learning algorithms due to their totally different starting points on the theory side as seen in Refs. [A1-A3]. Inspired by this idea of SVM, we propose to construct nonlinear mapping functions by arranging the data input into MZIs or cascaded MZIs networks, as shown in Fig. 1, in which the mapping schemes cannot be fully described by ANN.

SVM features better generalization capability and explicit mathematical formula (Ref. A4 reported an application comparison between ANN and SVM), while on the mathematical (or program) side, we cannot say which one is better or not since both are playing important roles in machine learning and statistics (Refs. [A1-A3]). However, if considering implementing them in photonics, there is an obvious

advantage for SVM compared to ANN. This SVM-like idea can be realized only by passive silicon photonics, without requiring nonlinear activation functions, while the performance upper limit will be determined by the SVM theory frame (Refs. [A1-A3]), varying according to applications. Thus, we have no additional cost for heterogenous integration. ANN cannot work well if without nonlinear activation functions. It is known that nonlinear functions are one of the main challenges in silicon photonics. Heterogenous integration is a possible solution, but optical nonlinear devices usually require high optical power for nonlinearity and optical amplifier devices also consume large power. Even though OEO devices or CPU can fit in, computing at light speed will be lost, and the claims of power advantages associated with utilizing photonics will become questionable. This explains the novelty of this work and our motivation to implement this idea in photonics, which has not been explored so far.

Fig. A1. Schematic of mapping functions: (a) quadratic function used in SVM. (b) sinusoidal function which can be implemented by photonic devices as phase-to-amplitude conversion.

(2) What difference: SVM vs ANN?

Here we clarify their difference in nonlinear classification principle using examples. An “ANN” used in this work means a multilayer perceptron (*i.e.*, neurons) with activation function f , which performs classification with a formula like $y = f(wf(\dots wf(wx+b)+b))$. For SVM, the nonlinear classification is to construct a hyperplane with a maximum margin in the feature space by utilizing the mapping function and in computer program it adopts kernel method to decrease the cost in calculating the dot product in feature space, like $y = w \langle G(x), G(v) \rangle + b \rightarrow y = wK(x,v) + b$. We did not directly use the kernel method in training the photonic device but construct a mapping function $G(x)$. Once the training is done, it is equivalent to the kernel method as seen below.

Next, we use the XOR example to prove that the projection in the photonic device satisfies the SVM conditions. Three simple $2 \times 4 \times 2$ ANNs are prepared in Fig. A2(a)-(c) to show how XOR can or cannot be separated by ANN (Source codes are in Suppl. Section 9). The programs in Figs. A2(d) and A2(e) emulate the photonic device. The bit pattern x is projected to x' in the complex space by two 2×2 MZI's phase-to-amplitude conversion. For simpleness, each MZI is assumed to be input with an optical vector $(1,0)$. We can figure out a linear equation in complex space by training the optical power. w and b can be implemented by photonic circuits. Then, we can rewrite the established $wx^T + b = y$ to $wx^T + b = 0$ by changing $x' \leftarrow [x', -y]$ and $w \leftarrow [w, [1,1]^T]$, which means a hyperplane in 5-dimensional complex space for one column of y_i (each column corresponds to a plane that does a one-vs-other classifier). The samples can be expressed as $x_d = [x', [0,1,1,0]]$ and then the distance $d(x, P)$ from x_d to the plane $wx^T + b = 0$ can be calculated as:

$$d(x_d, P) = \frac{|wx_d + b|}{\|w\|} = \frac{1}{\|w\|}$$

$$|wx_d + b| = \left| (-0.678 - 0.215i, -0.381 + 0.116i, -0.579 - 0.526i, -0.095 + 0.211i, 1) \begin{pmatrix} 0 & i & 0 & i & 0 \\ -1 & 0 & -1 & 0 & 1 \\ -1 & 0 & 0 & i & 1 \\ -1 & 0 & 0 & -1 & 0 \end{pmatrix}^T + (-0.476 - 0.126i) \right| = (1 \ 1 \ 1 \ 1), \text{ for column 1 of } y$$

$$|wx_d + b| = \left| (-0.63 - 0.189i, -0.570 + 0.046i, 0.463 - 0.228i, 0.990 - 0.159i, 1) \begin{pmatrix} 0 & i & 0 & i & 1 \\ -1 & 0 & -1 & 0 & 0 \\ -1 & 0 & 0 & i & 0 \\ -1 & 0 & 0 & -1 & 1 \end{pmatrix}^T + (-0.142 - 0.410i) \right| = (1 \ 1 \ 1 \ 1), \text{ for column 2 of } y$$

Thus, all data points have the same distances of $1/||w||$ to the plane, which indicates that the margin is maximized, and all bit patterns are support vectors. Training the power of output vector y essentially is same as training the distance in complex space and training the power of y (as done in Fig. A2[d]) is equivalent to minimizing $||w||^2$, which essentially is the same optimization target as used in conventional SVM algorithms (as shown by Eq. 1.24 in Ref. [A1]).

This photonic device can also be described by kernel method based on the dot product in the projected space. Once trained, the device can be expressed as $y = \alpha K(v, v') + \beta$, using a kernel function as defined in Eq. A1, where v and v' are two vectors in the projected space, the star * notes conjugate transpose. Then the decision function in SVM usually is written as $y = \sum \alpha_i K(v, v_i) + \beta$. As seen in Eq. A1, K can be derived to a form based on the Hermitian dot product of the projection function G (Eq. A2) which corresponds to x' in Fig. A2(d). (In general, G is not limited to Eq. A2 and can be constructed to other forms such as $w e^{in v}$ ($n = \text{integer}$) or $w e^{i \sum v_j}$ by inputting data to MZI networks, for which we can similarly derive a corresponding K). Then, we can obtain the K matrix (which is Gram matrix) from the dot product between each two samples as seen in Eq. A3.

$$K(v, v') = |\langle w_K, e^{i(v-v')} \rangle|^2 = |\langle \sqrt{w_K} e^{iv}, (\sqrt{w_K}^* e^{iv'})^* \rangle|^2 = |\langle G, G^* \rangle|^2 \quad (\text{Eq. A1})$$

$$G(v) = w e^{iv} \quad (\text{abbr. from Eq. 2 in the main text}) \quad (\text{Eq. A2})$$

$$K = \begin{pmatrix} 4110 \\ 1401 \\ 1041 \\ 0114 \end{pmatrix}, \quad \alpha = \begin{pmatrix} -0.5 \\ 0.5 \\ 0.5 \\ -0.5 \end{pmatrix} \quad \beta = 0 \quad (\text{Eq. A3})$$

It is easy to find α as shown in Eq. A3 to make $y = (-1, 1, 1, -1)$, indicating easy separation according to the sign or values (e.g., using $\beta=1$). In this way, the linearly inseparable data (as seen in a linear ANN in Fig. A2(c)) can be separated by utilizing projection. This projection-enabled separation is the core of SVM algorithm. As seen, the projection in the photonic device is equivalent to constructing a kernel function by which classification can be done in a similar optimization procedure as SVM.

Therefore, based on above discussion, we describe our device as an SVM-like classifier. Other mapping functions (in Fig. 1) can also be created to deal with more complex problems (Iris, Circle, Moon, Spiral, MNIST). As seen in the answer of Comment 4 below and Fig. 6 in the main text, this projection-based method can classify the nonlinear datasets with much less parameters than ANN and without activation functions, evidencing its difference from ANN since ANN will not work on these

highly nonlinear datasets if without activation functions.

(For simplicity, we use PPC to stand for the projection-based photonic classifier below)

Fig. A2. Three examples of simple $2 \times 4 \times 2$ ANNs for XOR classification: (a) purely linear without activation functions, (b) with ReLU, (c) with Sigmoid. Projection-enabled XOR separation using the phase-amplitude nonlinearity in MZI to emulate the photonic device: (d) π and (e) $\pi/2$ for the phase of bit 1. (f) Training accuracy vs epoch of the ANN in (c) and the projection-based programs in (d) and (e). (see Source codes in Suppl. Section 9)

[A1]: Schölkopf, B. & Smola, A. J. Learning with Kernels: Support Vector Machines, Regularization, Optimization, and Beyond Ch. 1&2 (MIT Press, London, 2002).

[A2]: Steinwart, I. & Christmann, A. Support vector machines: Information Science and Statistics Series (Springer, 2008).

[A3]: Schölkopf, B. & Smola, A. J. A tutorial on support vector regression, *Statistics and Computing*, vol. 14, pp. 199-222 (2004).

[A4]: Ren, J. ANN vs SVM: which one performs better in classification of MCCs in mammogram imaging, *Knowledge-Based Systems*, vol. 26, pp.144-153 (2012).

Comment 2:

Scalability: While I understand that the datasets that can be tested on this chip are always small, at least one simulation of a larger chip should be performed to demonstrate the whether the results of Fig. 5 scale to reasonably size machine learning problems of interest. SVMs are generally known to train faster than neural networks and are parametrized by only a relatively small number of support vectors in the datasets. For example, a similar comparison (multi-layer ANN vs nonlinear projection) might work on a dataset like MNIST. You might try Fourier-MNIST to keep the dataset to a reasonable size and try to limit the number of parameters and/or layers (see Williamson, et al., 2020 (cited) and Pai et al., 2020 "Parallel programming..." for a

two-layer approach). I would say this (or some similar argument) is required for publication, i.e. to complete the argument of scalability in the Discussion section.

Answer 2: Following your suggestions, we did MNIST simulation to examine the scalability of accuracy and power quantitatively, by gradually increasing the total parameters within the achievable range on our silicon photonic platform. Details are provided in the Suppl. Section 6 and Discussion on scalability is shown at Page 14, Paragraph 2 in the main text.

Here we directly give the scalability and classification results.

(1) Scalability. The accuracy scalability and power scalability were investigated. As shown in Fig. A3(a), the accuracy follows $\log(N^{0.13})$ for training and $\log(N^{0.09})$ for testing (N is the total parameter number). We sum up the trained phases at each phase shifter to calculate the total phase that is corresponding to the total power. As shown in Fig. A3(b), the total phase linearly increases with increasing N . Assuming using the same thermo-optic phase shifters, the power is about 5 W for 1663 parameters ($\sim 94\%$ accuracy), which is still much lower than traditional CPU or GPU. Figs. A3(c) and A3(d) show the accuracy dependences on the MZI layer number of VMM (see Fig. 1f) for 16 and 32 input components, respectively. Once the layer number is sufficient for reaching the maximum accuracy, further increasing the layer number cannot enhance the maximum accuracy for VMM.

(2) Results. Using 815 parameters and 16 input components, $\sim 90\%$ testing accuracy is obtained, which benchmarks a $>90\%$ accuracy with <900 parameters, as compared to a previous work (Ref. [A5], *Williamson, et al., 2020*). Using 1663 parameters and 32 input components, $\sim 94\%$ testing accuracy is obtained. Benchmark comparison to Ref. [A6] (complex photonic neural network, *Zhang et al., Nat. Commun. 12, 457 (2021)*) is summarized in Suppl. Section 8. It is known that ANN needs activation functions to achieve $>90\%$ accuracies. For the pure linear ANN, the maximum is about 85% (Ref. [A5]). If the principle of our device were exactly same as ANN, it would not be possible to achieve such accuracies without using activation functions, evidencing the different principle of our device. We clarify in Suppl. Section 6 that implementing projection can contribute to $\sim 4\text{-}5\%$ accuracy enhancement. **This principle can offer accuracies comparable/slightly higher to/than those up to $\sim 93\%$ for 2 layers in Ref. [A5], 90.5 in [A6], $\sim 93.4\%$ in [A7], 90% in [A8], and lower than 98% in Ref. [A9] (Pai et al., 2020) that used 64 components input and adopted a calibration method named “parallel nullification”.**

Fig. A3. Scalability examination by MNIST simulation. (a) Scalability of accuracy. (b) Scalability of the power (total phase) assuming using thermo-optic phase shifter of the π -shift power $P_\pi = 15\text{mW}$. Four architectures G1-G4 are explained in Suppl. Section 6. Accuracy in relation to MZI column (layer) number of VMM (Clements' topology [A7]) for G1: (c) 16-components input, and (d) 32-components input extracted from 2D FFT.

[A5]: Williamson, I. A. D., Hughes, T. W., Minkov, M., Barlett, B., Pai, S. & Fan, S. Reprogrammable Electro-Optic Nonlinear Activation Functions for Optical Neural Networks. *IEEE J. Sel. Top. Quant. Elec.* 26, 7700412 (2020).

[A6]: Zhang, H. et al. An optical neural chip for implementing complex-valued neural network. *Nat. Commun.* 12, 457 (2021).

[A7]: Lin, X., Rivenson, Y., Yardimci, N. T., Veli, M., Luo, Y., Jarrahi, M. & Ozcan, A. All-optical machine learning using diffractive deep neural networks. *Science* 361, 1004-1008 (2018).

[A8]: Xu, X. Y. et al. 11 TOPS photonic convolutional accelerator for optical neural networks. *Nature* 589, 44-51 (2021).

[A9]: Pai, S., Williamson, I. A. D., Hughes, T. W., Minkov, M., Solgaard, O., Fan, S. & Miller, D. A. B. Parallel Programming of an Arbitrary Feedforward Photonic Network. *IEEE J. Sel. Top. Quant. Elec.* 26, 6100813 (2020).

Comment 3:

*Training: The demonstration of BFO training in a photonic chip is interesting and should be emphasized more; I know you have a previous paper on the theory but you provide a very useful demonstration where you assess the effect of phase noise on the training. You might bring some results (e.g. comparison of forward finite difference / RMS prop to BFO) from the Supplement into the Main paper. Some more results and visualization of the BFO process might also be warranted. I think this will help clarify also the novelty of this paper: that a **trainable** matrix-vector multiply can be implemented within a photonic network of MZIs. If the authors can find a title and reorganize the paper around that thesis, it would much be easier to follow and appreciate the contribution of the work.*

Answer 3: Thank you very much for your valuable suggestions. We would like to follow your suggestions and emphasize the on-chip BFO training inside photonic circuits. We did detailed

comparison between BFO and RMSprop (forward finite difference using RMSprop optimizer) for both on-chip training experiment and simulation. Accordingly, we revised the introduction at Pages 1-3, added discussions about experimental comparison in Results in the main text (Pages 7-10), measured robustness at Page 12, Paragraph 4, and simulation comparison in Suppl. Section 7 (also summarized in Fig. A4 below). Some videos (see Suppl. Section 9) were uploaded to show the BFO process.

The title has been changed to “On-chip bacterial foraging training in silicon photonic circuits for projection-enabled nonlinear classification”. The paper has been substantially reorganized to emphasize this point as you suggested, and the novelty of this work as explained in the Answer 1 and 10. Source codes were also uploaded (Suppl. Section 9).

As shown in the following (1), in most previous works, the photonic/optical parts only perform inference, while leave training to be done by a regular computer program. On-chip training remains less demonstrated. For switching to different tasks, new neural network models must be trained first on computer. If the training is always stucked to another computer program, the energy and speed advantages will be lost for training and the photonics cannot be standalone. This work demonstrates both direct on-chip training and inference in the photonic circuits. All was done via the photonic device, without pre-training neural network models on computer, without processor-assistant classification at the middle or output stages, and without pre-calibration for phase errors.

(1) Brief overview of training methods of previously reported photonic neural networks.

Here we would like to overview several previous publications.

- (a) In Ref. [A10] (Shen et al): **Experiment work**; Training was done by an off-chip neural network (NN) on computer. Then the parameters (pre-trained by program) were copied to the photonic device based on calibration and matrix decomposition. The photonic device played inference for the pre-trained NN program and CPU was used to simulate activation functions.
- (b) In Ref. [A7] (Lin et al), **Experiment work**; An off-chip NN on computer was needed to design diffractive cards. The photonic device played inference for the pre-trained NN program. The network (using cards) cannot be reprogrammed.
- (c) In Ref. [A11] (Hughs et al), **Simulation work**; Proposed a new on-chip training algorithm using optical back propagation with *in-situ* gradient measurement via local power monitoring. Optical error was required to input from the output side. For real experiment, stably coding optical error vector might be challenging, and on-chip local OE feedback circuits are required.
- (d) In Ref. [A9] (Pai et al), **Simulation work**; Proposed a “parallel nullification” calibration method which also needs off-chip optical backpropagation to be done on computer. Then the parameters (pre-trained by program) were copied to the photonic device based on calibration curve. This method also needs local power monitoring and as mentioned in this paper, it may be influenced by inner loss imbalance.
- (e) In Ref. [A8] (Xu et al), **Experimental work**; Fiber-based convolutional optical neural network based on time-wavelength multiplexing. Optical inference only, as mentioned in this paper, the neural network was trained offline electronically. Inline training is still challenging.

Thus, optic/photonic inference remains a common way for demonstrating ONN (except for (c)). Our work demonstrates on-chip training in silicon photonic circuits to realize reconfigurable training and

inference, in a standalone mode without assistance from NN programs.

(2) Comparison between BFO and RMSprop

Gradient-based algorithms are widely used in training neural network in computer program, where the backpropagation (BP) algorithm is well known as a highly efficient one because the gradient of weight can be directly calculated from the output value (activation functions also satisfy this condition). But we know that currently it is almost not practical for photonic devices to do optical error backpropagation because (1) updating weights in electronic domain from optical domain requires accurate local measurement of optical phase and amplitude on chip and (2) it is challenging to prepare the optical error vector in complex domain from the current output optical signal. Ideally this is possible with correct phase measurement and control, but in real experiment, this is almost impossible if considering coding the error by a laser and fiber system with phase fluctuations. Thus, the forward propagation (FP) [Ref. A10] algorithm (evaluating the gradient by two times propagation) is one practical way for on-chip training. The gradient is evaluated one time for one parameter, with all other parameters kept unchanged. It is well known that the gradient-based algorithms easily encounter gradient vanishing and exploding problems, especially for RNN, inducing failure in training. Avoiding local minima problem for BP or FP algorithms is another difficulty. Recently, we proposed a modified bacterial foraging optimization (BFO) algorithm in Ref. [A12] and aim at real-time on-chip training photonic devices for arbitrary functions. As we demonstrated, BFO (global optimization) is another practical method for on-chip training even though it is not so efficient as the BP algorithm. It can offer almost same or even better performance as/than FP in our experiment. More important, BFO offers higher success rates and lower MSE losses than the latter in our experiment because of its capability of global convergence. The reason lies in that in BFO the current state vector moves to the next along a randomly chosen direction in the vector dimension and all parameters are updated simultaneously along this random direction. It is stochastic in nature (see the video of BFO in optimizing a 10-dimension sphere function) and offers a possibility to jump out from local minima. The BFO source code is shown in Fig. A4 below and as you suggested, in Suppl. Section 7 (see Fig. A5), we compare BFO to RMSprop for XOR, Iris, and MNIST classification including both training and testing. Discussion is given in Suppl. Section 7.

BFO source code

```

def BFOEng(self, data, yt, loss, delta, B1, Jlast, step, iserr, op, opn):
    for nc in range(5): #10
        for sb in range(B1.size(0)):
            dv = torch.rand(B1.size(1))*2.0 - 1.0
            Bc = B1[sb,:].detach().clone()
            Bc = Bc + step * dv/torch.square(dv).sum().sqrt()
            CopyB2M(self, Bc)
            yo, loss, acc, y, op, opn = self(data, yt, iserr, op, opn)
            for m in range(20):
                if (loss < Jlast[sb]):
                    Jlast[sb] = loss.detach().clone()
                    Bc = Bc + step * dv/torch.square(dv).sum().sqrt()
                    CopyB2M(self, Bc)
                    yo, loss, acc, y, op, opn = self(data, yt, iserr, op, opn)
                else:
                    break
            Jlast[sb] = loss.detach().clone()
            B1[sb,:] = Bc.detach().clone()
        sort = torch.argsort(Jlast)
        B2 = B1.detach().clone()
        B1[0:int(B1.size(0)/2),:] = B2[sort[0:int(B1.size(0)/2)],:]
        B1[int(B1.size(0)/2):int(B1.size(0)),:] = B2[sort[int(B1.size(0)/2):],:]
        Jtemp = Jlast.detach().clone()
        Jlast[0: int(Jlast.size(0)/2)] = Jtemp[sort[0:int(Jlast.size(0)/2)]]
        Jlast[int(Jlast.size(0)/2):int(Jlast.size(0))] = Jtemp[sort[int(Jlast.size(0)/2):]]

```

Fig. A4. BFO source code used in our experiment and simulation.

Fig. A5. Comparison on classification performance between BFO and RMSprop algorithms: (a) XOR, (b) Iris dataset, (c) Iris dataset with drop out, (d) MNIST dataset (5000 samples in total, 3000 for training, 2000 for testing). This is the case of 271 parameters in Fig. A3(a).

[A10]: Shen, Y. et al. Deep learning with coherent nanophotonic circuit. *Nat. Photon.* 11, 441–446 (2017).

[A11]: Hughs, T. W., Minkov, M., Shi, Y. & Fan, S. Training of photonic neural networks through in situ backpropagation and gradient measurement. *Optica* 5, 864–871 (2018).

[A12]: Cong, G., Yamamoto, N., Inoue, T., Okano, M., Maegami, Y., Ohono, M. & Yamada, K. Arbitrary reconfiguration of universal silicon photonic circuits by bacteria foraging algorithm to achieve reconfigurable photonic digital-to-analog conversion. *Opt. Express* 27, 24914–24922 (2019).

Comment 4:

Validity of statistical analysis:

Generally, the machine learning approaches should always have both training and testing/verification dataset curves to account for possible overfitting. For the plots in Fig. 5 in particular, I think this should be done.

Answer 4: We have changed Fig. 5 to a new figure (now Fig. 6), using more difficult Circle, Moon, and Spiral datasets. Both training and testing accuracies are provided. Details is added in the revised manuscript at Page 10, Paragraph 2.

Fig. A6 is prepared here for a quick look. The photonic device (PPC) is compared with 3-layer ANN models using ReLU activation functions and Softmax output layer. The same optimizer RMSprop with same learning rates is used for both ANN and the photonic device. We trained the photonic device with only 100 epochs, while training the ANN up to 500 epochs. As seen, the training and testing accuracies are almost same for each dataset, indicating no obvious overfitting. In comparison to ANN, PPC only has 43 parameters, but it gives almost same classification performance as ANN with >200 parameters. Meanwhile, PPC converges much faster with fewer training epochs. It is known that ANN can classify such nonlinear datasets because it uses nonlinear activation functions. However, PPC does classification for these datasets without nonlinear activation functions, manifesting its difference from ANN in classification principle. As we explained in the manuscript at Page 10, Paragraph 2 and the Answer 1, this nonlinear classification is owed to the constructed mapping function $G(x) = \{e^{2ix1}, e^{ix1}, e^{2ix2}, e^{ix2}\}$. Therefore, the result can differentiate the PPC from all previous photonic ANNs and evidence the role of projection in nonlinear classification.

Fig. A6. Comparison between ANN and the projection-based photonic classifier (PPC) on determining nonlinear boundaries for the circle, Moon, and Spiral datasets. For Moon and Circle, 1000 samples in total, 600 for training, 400 for testing; for Spiral, 1500 samples in total, 900 for training, 600 for testing. Accuracies for both the train and test sets are given. The ANN uses the Relu activation functions and Softmax output layer. (Source codes for simulating both ANN and PPC were uploaded, see Suppl. Section 9)

Comment 5:

Reproducibility:

The experiment appears to be reproducible and in line with other implementations, but I have a question about an experimental detail. Thermal control is typically required to ensure that the heat transfer throughout the chip doesn't introduce any spurious phase shifts throughout the device. It is unclear if this is part of the setup and whether it affects training. The authors should elaborate on this point.

Answer 5: During this time, we packaged another chip and did wire bonding. With this wire bonding chip, reproducibility, robustness, and long-time stability have been systematically studied and verified with adding more quantitative data. The corresponding results are added in the Discussion in the main text (at Page 12, Paragraph 4 ~ Page 14, Paragraph 1) and Suppl. Sections 5 and 4.4.

Here for your quick reference, I would like to briefly reply this comment from the following three aspects. **(1) Reproducibility and stability.** Using this wire-bonding chip, we repeated all training experiments, showing sample reproducibility. We also repeated the training for many times and confirmed the training reproducibility. The wire-bonding chip shows much superior long-time stability than the chip using 40-pin probes. Fig. S16 in Suppl. Section 5.1 shows the long-time stability of Iris classification performance. For each port, the long-time control reproducibility was also checked. We first use BFO to obtain the voltage setup to maximize the power at each port and then check the MSE error change within one week. As seen in Fig. A7, the error almost keeps at the same low level as just trained, showing long-time stability and control reproducibility even without any special thermal control under our experimental conditions. **(2) Why spurious phase shifts are negligible.** Both our previous and current experiment were done at room temperature without any special thermal management as seen in Fig. A7. The chips also have no special thermal sink design. The temperature around the heater is estimated to be $\sim 120^\circ\text{C}$ for π -phase shift. Since the surface passivation oxide layer is very thin, just 200 nm, much thinner than that underside thickness ($>4\ \mu\text{m}$) and in-plane heater distance (100 μm to the nearest neighbor), the power balance (electrical power vs heat flux out) is mainly established at the surface locally around the heater. Thus, the spurious thermal interference between heaters is negligible, which has been proved in our large-scale switches of high density (>2000 heaters) in Refs. [A13, A14]. Comparatively, the heater number is much smaller (<50 heaters) in this work. **(3) Understanding the robustness to device imperfections for classification application.** We also would like to discuss the influence of device imperfect (*e.g.*, from fabrication error) on reproducibility. It is well known that the most sensitive component is the directional coupler (dc). For switching application, the dc is required to be as close as to the ideal 3 dB to guarantee low crosstalk. To examine how the classification is influenced by the dc error, we introduce the dc deviation δ as seen in the equation in Fig. A8 which means the deviation ratio from $\pi/4$ (an ideal dc) and is randomly generated for all dc in the device. Taking XOR as the example, as shown in Fig. A8, even with an error up to 10%, the final MSE in training is almost same as that without error. The training is still successful even with dc errors. As seen from the optical propagation inside the device, almost same separation result can be trained out. This is different from switching application, because for the switches, the light always goes along a single path that is greatly influenced by the dc error in each MZI along the path; but for the classification, the light experiences multiple path interference and even with the dc

errors, the training will re-figure out a different multiple path interference that can convey the same classification information at the output. This can be understood from the light propagation in Fig. A8. Therefore, this classification device essentially is more robust to fabrication error and thus has higher reproducibility than other silicon photonic devices using directional couplers such as switches.

Fig. A7. Experimental setup, chip after package, long-time stability, and cross section figure of the device.

Fig. A8. XOR separation with and without directional coupler (dc) deviation δ . The mapping figures shows the simulated optical power propagation inside the device (a) without dc error and (b) with dc error. The definition of a deviation δ is shown in the equation. (c) Comparison on MSE with and without including δ for XOR and Iris.

[A13] Tanizawa, K. et al. Ultra-compact 32x32 strictly-non-blocking Si-wire optical switch with fan-out LGA interposer. Opt. Express 23, 17599-17606 (2015).

[A14] Suzuki, K. et al. Strictly Non-Blocking 8x8 Silicon Photonics Switch Operating in the O-Band. J. Lightw. Tech 39, 1096-1101 (2021).

Comment 6:

Organization:

The paper lacks some impact structure and therefore struggles to bring home the core message and novelty of the work. Mainly, I have to do a lot of work to figure out whether this work really adds any value above what is already in the field. For example, after the results are discussed, there is some valuable analysis, but there is no discussion to compare those results to known benchmarks in the field (Shen et al. 2017, Zhang et al. 2021). There is also insufficient discussion of the core differences of SVMs and ANNs as it relates to real-world applications; it seems hand-wavy in the Supplement's Scalability section with little data/experiments to back it up.

Answer 6: We have responded to this comment by thoroughly revising and reorganizing the manuscript with more data added. I think we have explained in detail in the Answers to other comments above and below for your concerns here, which are also clarified in the revised manuscript and Suppl. file.

For distinguishing with other previous papers, please refer to the Answer 3 and Introduction in the revised manuscript.

For the core difference between SVM and ANN, please refer to the Answer 1, Suppl. Section 1, and the *principle* in the main text (at Page 4, Paragraph 2 – Page 7, Paragraph 2).

For scalability, please refer to the Answer 2 above, Suppl. Section 6, and Discussion at Page 14, Paragraph 2.

For Supplementary file, this file has been rewritten with more data and detailed discussions added.

For benchmark comparison, comparison on all benchmarks with Ref. [A6] (Zhang et al., Nat. Commun. 12, 457 (2021)) is given in Suppl. Section 8. Comparison on MNIST is given in Answer 2 above, Page 14, Paragraph 2 in the main text, and Suppl. Section 6.

Both Shen et al 2017 and Zhang et al 2021 are not on-chip training. The paper (Shen et al 2017) did vowel classification that is not the target of this work, thus we compare all benchmarks with Ref. [A6] (Zhang et al., Nat. Commun. 12, 457 (2021)) that demonstrated similar benchmarks. For Iris, we achieved on-chip testing accuracy 98.3% and 96.7% (see Results in the main text at Page 9, Paragraph 3, and Suppl. Section 4), comparable to 97.4% in Ref. [A6] (Zhang et al). For Moon, Circle, we achieved 100% and >98% for 2-classes (see C10 in Suppl. Section 9) and ~90% 3-classes Spiral. For these traditional benchmarks, our work offers comparable performances to this paper and demonstrated novel functions such as combinational logic operation (half adder, etc) (at Page 9, Paragraph 2) in experiment that are not reported so far, as far as we know.

Comment 7:

Specific comments:

Figures:

Fig 5- is a bit too busy, please limit to fewer plots with bigger axes.

Fig 4- colormap should be changed to a non-rainbow colormap for clarity

Fig 3- The colors for each port are a bit distracting. You could make all the bars that are not part of the classification be a gray

color and highlight the bars we are supposed to pay attention to with a blue or red color. Alternatively, use a 2d colormap instead.

Fig 1- There is a strange redundancy in your architecture. If you have two MZIs next to each other, you can reparametrize the system as a single MZI. You can just flip the architecture so that doesn't happen. Also in case you wanted to show a universal 8x8 architecture in the figure,

Answer 7: Thank you very much for these kind comments. We have made corresponding revisions as you advised in the revised manuscript. To briefly summarize here as:

(1) About Fig. 5 (now Fig. 6), this figure is changed to a new one based on more difficult datasets (see the answer to Comment 4 above). Bigger axes are used.

(2) About Fig. 4 (now as Fig. 5), the colormap figures are changed to grayscale.

(3) About Fig. 3, this figure is changed to grayscale map as well.

(4) Fig. 1 has been revised. That two columns of MZIs next to each other are not redundant. Actually, a column of phase shifters exists there, which is used for adjusting the vector x' .

Comment 8:

Equations:

Re-define B in Eq 4 for clarity in terms of the terms in Eq 1

I do not fully understand Eq 4, if B is a matrix and H is a scalar inner product, then how does it match the shape of B ? There might be a notation issue here.

Answer 8: H is a matrix consisted of scalar conjugate inner product between each two samples x and v , not just one scalar. To be clear, equations (3) and (4) has been revised in the main text as shown below. When y_t is a matrix of (output node $M \times$ sample number N), H is a matrix of ($N \times N$), B will be a matrix of $M \times N$ that will be implicitly constructed automatically by training the device. (In answer 1, for simpleness, y_t is reduced to a vector, so does B . Then BH is equivalent to vector-matrix multiplication, as seen above in Answer 1 and Suppl. Section 1).

$$H_{x,v} = \langle G(x), G(v)^* \rangle = \left[\sum_j (|w_j|^2 e^{i(x_j - v_j)}) \right], \sum |w_j|^2 = 1 \quad (3)$$

$$\mathcal{L} = \min \sum (y_t - |BH + b|^2)^2 / N \quad (4)$$

Comment 9:

Other:

The authors mention some alternatives to heaters for phase shifters. I urge reading work on MEMS phase shifters and phase change materials which are both low-loss and take up no power in static configuration. A sentence or two should cover that in the Discussion.

Answer 9: As you suggested, we add the following references Refs. [A15-A18] (MEMS in Ref. [A15] and phase change materials (PCM) in Refs. [A16-A18]) for alternatives of phase shifters. In discussion, we have added "If other phase shifters with high energy efficiency such as *pin*-type^{36,37}, PCM-type^{8,38}, or MEMS-type⁴⁰ are used to replace the heaters, the total power can be further decreased. The *pin*-type phase shifter using carrier injection usually has a P_π about 2-3 mW, The PCM and MEMS phase shifters can implement VMM with low loss and powerless standby states." at Page 15, Paragraph 1.

[A15] T. J. Seok, et al. Wafer-scale silicon photonic switches beyond die size limit. *Optica* 6, 490-494 (2019).

[A16] J. Feldmann, et al. Parallel convolutional processing using an integrated photonic tensor core. *Nature* 589, 52-58 (2021).

[A17] C. Wu, et al. Programmable phase-change metasurfaces on waveguides for multimode photonic convolutional neural network. *Nature Communications* 12, 96 (2021).

[A18] B. J. Shastri, et al. Photonics for artificial intelligence and neuromorphic computing. *Nature Photonics* 15, 102-114 (2021).

Comment 10:

Summary:

Publication of this paper depends on differentiating what is now a common photonic architecture (MZI networks) from existing implementations. The authors attempt to do this by framing the architecture as a nonlinear SVM kernel implementation that can be trained from scratch, but I am not yet convinced that it reaches the necessary level of impact for publication in Nature Communications. We already know these chips can do matrix multiplication, so applying it to an SVM (which is basically just another ANN, or really any machine learning model with some matrix-multiplication) doesn't seem impactful enough on its own, and free space implementations seem to do much better by scaling the SVM to much larger matrix sizes than can be achieved on a chip. Therefore I cannot recommend publication in the current form, but perhaps with significant revision I could be convinced otherwise. The authors should emphasize the more impactful training results, perform more simulations on the SVM architecture (applied to bigger multi-dimensional datasets like MNIST), and reorganize the paper to more clearly lay out the advantages of the overall approach over existing neural network architectures (this should be in the main paper, not supplement).

That said, it is an impressive feat whenever a group manages to implement a matrix-vector product on a photonic chip. Any optics journal would be suitable to publish this work as is if the authors want to stick with SVM/nonlinear projection as the main contribution.

Answer 10: Thank you very much for so many enlightening comments, which greatly help us to improve our paper. I think we have clearly explained why we describe this device as SVM instead of ANN in the Answer 1 above as well as in the main text and Suppl. Section 1. The core value of this work has two points that (1) we demonstrate on-chip bacterial foraging training in silicon photonic circuits without relying on pre-trained neural network models on computer, which can be used for various real-time training purposes for photonic devices; and (2) propose to implement the projection-based classification principle by constructing nonlinear mapping functions in photonic circuits, which offers a possibility for implementing machine learning in photonics without leveraging nonlinear optical effects. Our results show that it is possible to achieve easy classification with much fewer parameters and faster convergence if a suitable projection function can be created on chip. These two points can differentiate this work from all previous works done in integrated photonics. In addition, there are many possibilities to construct useful mapping functions by following our proposal, which could ease classification tasks or have some other potential applications (such as data preprocessing).

As mentioned in above answers, the evidence to support these two points includes XOR, nonlinear datasets, and MNIST, for which ANN cannot get that high accuracies if without activation functions. But

in this work, we do not intend to deny ANN which is a notable general algorithm to machine learning, while as a special algorithm, this SVM-like one also has many potential applications and flexibility.

To the last, we would like to give some comments on different optical forms to implement machine learning. Strong comparison between them from mono-aspect is not fair since each has pros and cons, really depending what the application scenario is.

Free space implementations are a good scheme to implement optical neural network or random projection, especially for treating 2D images. It can be easily scaled to extend the matrix dimension in 2D. Reprogramming and optical training are the challenges. A large-scale CMOS sensor may be also required to save images for final classification by external processors.

For integrated photonics, the advantages include re-programmability, integration degree, reliability, and flexibility of integration with optical communication system. But the challenges are scalability, nonlinear functions, and efficient on-chip training. It may be difficult to reach that large scale on a single chip, but the large-scale optical systems can be made by optical interconnects and utilizing multiple wavelengths. The device based on projection can also fit in this interconnect scenario with various possibilities for computing, classification, and electrical signal processing by optical means, which is one direction in our future study. If applying the photonics to data pre-processors with allowing assistant external electronics, constructing a projection function in photonic circuits could greatly simplify the behind electronics, hence being capable of lowering the system cost and power.

Note that source codes related to the data used in this response letter can be referred to in Suppl. Section 9.

REVIEWER COMMENTS

Reviewer #2 (Remarks to the Author):

Based on the rewritten paper, I think the novelty of the paper is much clearer. In particular, it seems to me that the paper's main contribution is in the on-chip training, and I now agree that the SVM nonlinear projection used in this work help to distinguish it from other newly proposed training approaches such as genetic algorithms.

At this point, most of my issues with the paper come down to clarity and language. More clear notation is still recommended for explaining the SVM architecture in a field saturated with optical ANNs. Overall, I find the notation and the methods for the SVM very confusing and need a lot of effort to understand it. More background explaining support vectors / SVMs more generally would be recommended in the main text (I am familiar with SVMs and yet still do not fully understand how inputs are constructed based on reading the text).

My suggestion is to define the SVM inputs clearly with examples to motivate it (e.g. Iris and XOR), devoting a section to explain this. Then, in a separate section, define the transformation implemented by the optical device. Importantly, this is an excellent time to explain the advantage of an SVM "phase encoding" approach compared to the ANN approach. A figure (to replace Fig 1, which is very unclear to me) to show the improvements of an SVM over an ANN is then recommended (this helps guide the reader on the novelty of the application). All of this should be in the main text. It would also help to compare the equations of an ANN and SVM, possibly in table format (inputs, device operator, error/cost/loss function).

Finally, I do think it is inaccurate to say that phase measurement at the output of the circuit is difficult. Zhang et al. 2021 have achieved this through interference at the end of the circuit. Phase fluctuations in the laser don't matter since it is a global phase and we use self-interference which ignores global phase. So this still puts gradient-based measurement / backpropagation at play in on-chip training demonstrations, given the ability to send light forwards and backwards through the circuit and measure at various points in the circuit (both of which are certainly feasible, though perhaps less scalable).

I think most of my concerns in the previous review have been successfully addressed and think this would be suitable for publication after the next cycle. Nice work!

Responses to the reviewer (for the 2nd review)

We have fully considered the reviewer's comments and suggestions and correspondingly made additional revisions to our manuscript to improve the clarity. The main revised parts were highlighted in yellow color in the redline version of the manuscript (a clean version of the manuscript without color highlighting was also uploaded).

In this response letter, all the reviewer's comments are addressed below in a reproduced verbatim style (the comments are cited in *Italic* format and answered in a point-by-point style).

Comment 1:

Reviewer #2 (Remarks to the Author):

Based on the rewritten paper, I think the novelty of the paper is much clearer. In particular, it seems to me that the paper's main contribution is in the on-chip training, and I now agree that the SVM nonlinear projection used in this work help to distinguish it from other newly proposed training approaches such as genetic algorithms.

Answer 1: Thank you very much for your valuable comments and suggestions, which are very enlightening and greatly help us to improve our manuscript. Also, your recognition for our work is highly appreciated.

Comment 2:

At this point, most of my issues with the paper come down to clarity and language. More clear notation is still recommended for explaining the SVM architecture in a field saturated with optical ANNs. Overall, I find the notation and the methods for the SVM very confusing and need a lot of effort to understand it. More background explaining support vectors / SVMs more generally would be recommended in the main text (I am familiar with SVMs and yet still do not fully understand how inputs are constructed based on reading the text).

Answer 2: (1) To explain the SVM architecture more clearly and explain how the inputs are constructed, we made revisions by following your suggestions in Comment 3. Notations and the differences between our SVM-based method and ONN have been clarified. Please see the corresponding revisions explained in Answer 3 below. (2) To explain the background and projection-based idea of SVM, we add some more general descriptions on SVM at Page 3, Paragraph 2 and at Page 5, Paragraph 2, Line 1-5. (3) In addition, all languages have been examined and polished.

Comment 3:

My suggestion is to define the SVM inputs clearly with examples to motivate it (e.g. Iris and XOR), devoting a section to explain this. Then, in a separate section, define the transformation implemented by the optical device. Importantly, this is an excellent time to explain the advantage of an SVM "phase encoding" approach compared to the ANN approach. A figure (to replace Fig 1, which is very unclear to me) to show the improvements of an SVM over an ANN is then recommended (this helps guide the reader on the novelty of the application). All of this should be in the main text. It would also help to compare the equations of an ANN and SVM, possibly in table format (inputs, device operator, error/cost/loss function).

Answer 3: Thank you very much for your valuable suggestions. We would like to follow your suggestions and made the revisions: (1) A new section "Data input and device operators" was added to clearly show data input and transformation for XOR, Iris, and nonlinear datasets. The mapping

function $G(x)$ was written exactly. Accordingly, Fig. 1 was changed to a new figure to clearly define the inputs, based on which a series of equations are given to show the mapping functions. (2) Another section “Model explanation” was also added to explain the SVM-like principle. A Table 1 (as seen below) was prepared to summarize the difference between ONN and our PPC. As you suggested, all these parts are added into the main text.

Table 1 Model comparison between ONN and PPC in this work

	Algorithm	Equation	Data input				Device operator	Activation function f	Loss function	Refs.	Optical bias port for coded data?
			Domain	Coded to	After coding	trained?					
ONN	ANN	$y=f(\dots f(Wf(Wx)))$	Optical	Amplitude	x	no	Linear operator W (matrix)	$w/$	same	[6,14,15,17]	$w/ [14, 15]$
				Intensity						[7,8]	
PPC	SVM	$y=WG(x)$	Electrical	phase	$G(x)$	yes	Nonlinear projection operator G and linear operator W	w/o		this work	w/o

The main revisions are located at Page 5, Paragraph 2; a new section “Data input and device operators” at Page 6, Paragraph 2; and a new section “Model explanation” at Page 8, Paragraph 1 to Page 10, Paragraph 2. The description on Table 1 is at Page 10, Paragraph 1.

Comment 4:

Finally, I do think it is inaccurate to say that phase measurement at the output of the circuit is difficult. Zhang et al. 2021 have achieved this through interference at the end of the circuit. Phase fluctuations in the laser don't matter since it is a global phase and we use self-interference which ignores global phase. So this still puts gradient-based measurement / backpropagation at play in on-chip training demonstrations, given the ability to send light forwards and backwards through the circuit and measure at various points in the circuit (both of which are certainly feasible, though perhaps less scalable).

I think most of my concerns in the previous review have been successfully addressed and think this would be suitable for publication after the next cycle. Nice work!

Answer 4: Thank you very much for pointing out my inaccurate description on the phase measurement and I am sorry for the inappropriate expressions. Actually, I totally agree with you that both the coherent detection and light backpropagation are feasible, provided taking additional efforts such as integrating monitors at various points for gradient measurement as you mentioned and amplifier circuits for local OE conversion. Thus, we made the below revisions accordingly.

1. I think the most related sentence to your comment locate at the Page 10, Paragraph 2, Line 4 in the previous response letter: “*But we know that ... with phase fluctuations.*” I would like to revise this sentence to “*Since on-chip monitors and coherent detection are achievable, optical backpropagation is also feasible [Ref. 15], provided taking additional efforts such as local monitors for gradient measurement, amplifier circuits for local optical-to-electrical feedback, and preparing the optical feedback vector.*”
2. In main text, at Page 2, Paragraph 1, Line 21, we revised to “*In theory, BP-based optical training^{15,17} and offline-calculation-assisted parallel calibration methods¹⁸ were proposed, which are feasible provided taking additional measures like utilizing locally embedded monitors and amplifier circuits for local optical-to-electrical feedback and preparing optical feedback vectors. Thus, applying the weights trained by an offline model to pre-calibrated photonic devices directly or via matrix decomposition remains a mainstream approach to demonstrate an inference-only ONN in experiment.*”

Finally, we would like to let you know that we have revised “Code Availability” and Suppl. Section 9.

The source codes we uploaded can be used for review purpose and kept permanently in the submission system, but the codes developed in our institute have the IP license and cannot be directly open to the public download. Due to this issue, therefore, we would like to revise “Code Availability” as “The code that supports the findings of this study are available from the corresponding author upon reasonable request”. In addition, Suppl. Section 9 was also revised accordingly, but the source code of BFO engine is remained in Fig. S26.

REVIEWERS' COMMENTS

Reviewer #2 (Remarks to the Author):

All comments appear to be addressed. Should be good to go from my end!